# Jagged-mediated lateral induction patterns Notch3 signaling within adult neural stem cell populations

Sara Ortica ⓘ , Miguel Martinez Herrera ⓘ , Louis Degroux, Bastian Rochette, Nicolas Dray ⓘ ✉ & Laure Bally-Cuif ⓘ ✉

In the adult brain, Notch3 signaling promotes neural stem cell (NSC) quiescence and stemness. It remains unknown how Notch3 signaling levels are controlled and relate to these NSC decisions. Here we directly measure the nuclear translocation of the Notch3 intracellular fragment (N3ICD) and quantify Notch3 signaling in NSCs of the zebrafish adult telencephalon in situ. We report that Notch3 signaling levels match NSC quiescence and stemness levels. In physical space, Notch3 signaling is patterned and high signaling levels surround N3ICD[low] cells, which also express the *deltaA (dla)* ligand. Another ligand, *jagged1b* (*jag1b*), expressed in all NSCs, activates Notch3 signaling and sustains expression of the stemness factor Sox2. Finally, lowering *jag1b* preserves the structured distribution of Notch3 signaling levels in space but attenuates their variance. We propose that Notch3 signaling integrates Dla-mediated lateral inhibition and Jag1b-mediated lateral induction to control quiescence and stemness and their spatiotemporal dynamics in adult NSCs.

Neural stem cells (NSCs) generate neurons in the adult vertebrate brain, contributing to brain plasticity, repair, and growth. NSC potential ("stemness") follows individual NSC decisions such as quiescence exit, self-renewal or neurogenesis commitment, and is also tightly regulated temporally and spatially across the niche at the population scale[1,2]. Numerous regulators of individual NSC activity have been identified, but it remains largely unknown how the spatiotemporal nature of stemness is encoded.

In the major adult NSC niches (sub-ependymal zone of the lateral ventricle (SEZ) and sub-granular zone of the dentate gyrus (SGZ) in mouse, ventricular zone of the pallium in zebrafish[3–5]), NSCs are quiescent radial glia-like cells (RGs). They activate to divide and self-renew or generate further committed neural progenitors (NPs) fated to delamination and neuronal differentiation[6–13]. The Notch signaling pathway is a major regulator of NSC decisions along this lineage, in a complex receptor-dependent and context-dependent manner[13–15]. Several Notch receptors are expressed in NSCs (Notch1, 2 and 3 in mouse[16–22], and Notch1a, 1b and 3 in zebrafish[23–25]). Notch1 expression and signaling are predominant after NSC activation to limit neuronal

differentiation[16,24]. In contrast, Notch2 and 3 are strongly expressed in quiescent NSCs and promote quiescence: upon lowering Notch3 activity in zebrafish, quiescent pallial NSCs enter proliferation[24]; likewise, abrogation of Notch2 and Notch3 also leads to NSC quiescence exit and proliferation in the mouse SEZ and SGZ[20–22]. In the zebrafish pallium, Notch3 signaling also maintains the NSC progenitor state, likely via a distinct downstream pathway involving the transcription factor Hey1[26]. Finally, in physical space, Notch signaling also influences the spatial distribution of NSC activation decisions: in the zebrafish pallium, NPs prevent NSC activation in their vicinity, and this is abolished when Notch signaling is abrogated[27]. Together, these results identify Notch signaling, in particular Notch3, as a major regulator of the spatiotemporal aspects of stemness, contributing to controlling quiescence, progenitor properties, and the proper distribution of NSC decisions in physical space.

The challenge now is to understand Notch3 signaling in situ in quantitative, dynamic and mechanistic terms across adult NSC niches. Upon binding a ligand (Delta or Jagged) expressed by a contacting cell, a major output of Notch signaling in the receiving cell is

Institut Pasteur, Université Paris Cité, CNRS UMR3738, Zebrafish Neurogenetics Unit, F-75015 Paris, France. ✉e-mail: nicolas.dray@pasteur.fr; laure.bally-cuif@pasteur.fr

transcriptional[14,28,29]: Notch cleavage by the gamma-secretase complex releases its intracellular fragment (NICD), which translocates to the nucleus where it binds the cofactor RBPj to activate target genes. This is the case in adult NSCs, where NICD overexpression rescues Notch signaling blockade to promote quiescence[22,23], and Notch invalidations generate phenotypes[16,21,22,30] that resemble RBPj invalidations[31–33]. Canonical Notch signaling is a direct pathway that does not involve second messengers[34,35], and quantitative measures in Drosophila and C. elegans indicate that nuclear NICD levels correlate with the duration of transcriptional bursts in target genes, thus are quantitative indicators of Notch signaling potential[36–38] -although, depending on context, additional regulators may be needed[39]. We exploited these findings to design a direct tracer of Notch3 signaling activity in zebrafish adult pallial NSCs in situ. Inspired by pioneering work in Drosophila[40–42], we knocked in fluorophores into the C-terminal end of Notch3-ICD (N3ICD) and monitored N3ICD-fluo translocation to the NSC nucleus. We show how the generated lines allow a reliable quantification of nuclear N3ICD in NSCs in real time and in physical space across the NSC population, and use this tool to identify the ligands controlling Notch3 signaling in situ. Our results indicate that the Notch3 signaling pattern at the population level responds to both lateral inhibition and lateral induction, the latter being mediated by Jagged1b, expressed by most NSCs, and controlling stemness. Our results further support a role for Jag1b in potentiating lateral inhibition. These results reveal how the different ligand-receptor interaction modes permitted by Notch signaling are integrated to encode the spatiotemporal dynamics of Notch3 signaling and ensure quiescence and stemness within the adult pallial NSC niche.

## Results

### Generation of direct in vivo reporters of Notch3 expression

We designed two strategies to generate Notch3 signaling readout lines, which together provide versatile technical applications: to quantify nuclear N3ICD in transgenic backgrounds with various fluorophores, to track Notch3 in different subcellular compartments, and to test the effect of *notch3* gene dosage.

First, a BAC encompassing the entire *notch3* locus, plus 150 kb of upstream sequence, was recombined in *E. coli* to insert a *GFP-P2A-nlsRFP* cassette in position +6823 of the coding *notch3* region (Fig. 1A). This fuses GFP to Notch3 within N3ICD after amino acid 2274 (MLLHQ), upstream of the first PEST sequence (Supplementary Fig. S1A–C). nlsRFP will translocate to the nucleus and provide a nuclear marker for *notch3*-expressing cells, useful for nuclear segmentation and quantification. This line is referred to as TgBAC(*notch3:notch3-GFP-P2A-nlsRFP*)[ip13Tg], abbreviated in TgBAC[n3-GFP/+] (for heterozygote animals), and contains an intact endogenous *notch3* locus.

Second, we used Crispr/Cas9 and Homology Directed Repair (HDR) recombination to knock-in an *Azamigreen-P2A-nlsRFP* cassette in position +29155 of the endogenous *notch3* locus (Fig. 1B and Supplementary Fig. S1A, B). Azamigreen (AG) was chosen for: (i) its green fluorescence, combinable with nlsRFP in direct imaging, (ii) its detection by a specific anti-AG antibody, for combination with GFP lines by immunohistochemistry (IHC), and (iii) its different sensitivity to pH compared to GFP. The recombination site was selected upon comparing the efficiency of several gRNAs and led to a manageable efficiency of germline transmission (Supplementary Fig. S1D–F). It generates a fusion of AG to Notch3 within N3ICD after amino acid 2382, C-terminal to the first PEST sequence (GSTAG) (Supplementary Fig. S1B). This line is referred to as TgKI(*notch3:notch3-mAG-P2A-nlsRFP*)[ip14Tg], abbreviated in TgKI[n3-AG/+] (for heterozygotes) or TgKI[n3-AG/n3-AG] (for homozygotes).

We first verified that GFP and AG displayed the expected expression pattern in the embryonic and adult brain. In situ hybridization for *gfp* in TgBAC[n3-GFP/+] embryos, and for *azamigreen* in TgKI[n3-AG/+] embryos, revealed a pattern identical to *notch3* at 24 hours-post-fertilization

(hfp) (Supplementary Fig. S2A). Direct imaging of GFP and nlsRFP expression in the forebrain of 24hpf TgBAC[n3-GFP/+] embryos also showed nlsRFP confined to cell nuclei in neural progenitors, while GFP intensely labeled cell membranes and more weakly nuclei (Supplementary Fig. S2B). Finally, direct whole-mount imaging of adult brains at 3 months post-fertilization (mpf), with focus on pallial ventricular neural stem and progenitor cells (NSPCs), revealed a similar pattern for nlsRFP/GFP and nlsRFP/AG in TgBAC[n3-GFP/+] and TgKI[n3-AG/+] animals, with visible membrane and nuclear staining (Fig. 1C, D).

In addition, we observed fluorescence in numerous cytoplasmic dots in both lines, very motile in embryonic neural progenitors imaged dynamically (Supplementary Fig. S2C). Based on previous observations in Drosophila[42], we tested whether they correspond to the endocytic trafficking of Notch3-fluo. We compared the localization of Notch3-GFP and Notch3-AG with Rab proteins in early/sorting (Rab5[pos]), recycling (Rab11[pos]) or late (Rab7[pos]) endosomes, or the Lysosome-associated membrane protein 1 for lysosomes (Lamp1[pos]), in NSPCs. Expression of Zona occludens 1 (Zo1) was used to delineate the NSPC apical surface. We could detect colocalization of some Notch3-GFP with recycling endosomes and lysosomes (Supplementary Fig. S2D), while Notch3-AG, in addition, colocalized with early/sorting endosomes (Supplementary Fig. S2E). Thus, Notch3 is also engaged in dynamic endocytosis, recycling and degradation in adult pallial NSPCs.

### *notch3* expression in the adult pallium encompasses the entire NSC population

NSCs and NPs are intermingled at the pallial ventricular surface. NSCs express the radial glia marker Glial fibrillary acidic protein (Gfap), while NPs do not. Both cell types express the progenitor marker Sox2. To position Notch3 expression among NSPCs, we compared nlsRFP, Gfap and Sox2 using IHC (Fig. 1E–H). Among apical Sox2[pos] cells, over 96% of Gfap[pos] cells were nlsRFP[pos], and over 94% of nlsRFP[pos] cells appeared Gfap[pos] (Fig. 1G). This confirms *notch3* expression in NSCs. The proportion of strongly nlsRFP[pos] cells was 90%, with an additional 5% harboring low nlsRFP levels, some of which were Gfap[neg]. This also indicates the presence of nlsRFP in NPs (Fig. 1H), either due to the stability of nlsRFP or to low *notch3* expression extending to NPs. The similar pattern of nlsRFP in the two lines also confirms that the regulatory elements contained in TgBAC[n3-GFP/+] drive proper Notch3-GFP expression in the adult brain.

### Notch3 fusion proteins are functional

To assess the functionality of Notch3 fusion proteins, we used *notch3[fh332]* null mutants[24]. *notch3[fh332/fh332]* larvae display a "curved" phenotype at 7dpf[43] and generally die between 10 and 15dpf, while *notch3[fh332/+]* heterozygotes are morphologically normal and adult viable. Pallial RG in *notch3[fh332/fh332]* larvae also display decreased quiescence entry starting at 5dpf, in link with Notch3 promoting RG quiescence[26].

TgKI[n3-AG/n3-AG] homozygotes displayed no overt phenotype at the larval stage and survived normally until adulthood. Focusing on neurogenesis, we assessed the proliferation and neurogenic activity of embryonic and larval RG and adult NSPCs, in TgKI[n3-AG/n3-AG] fish. In embryos, RG proliferation rate was measured using a 1-hour BrdU pulse, then immediate IHC (Supplementary Fig. S3A, B), and neurogenesis with the neuronal protein HuC/D (Supplementary Fig. S3C–E). TgKI[n3-AG/n3-AG] embryos behaved identically to wild-type (WT) for these parameters (Supplementary Fig. S3B, D, E). In larvae, like for *notch3[fh332/fh332]* mutants[24], we assessed telencephalic RG quiescence in TgKI[n3-AG/n3-AG] homozygotes by quantifying the proportion of RG (Brain lipid binding protein-positive, Blbp[pos]) expressing the proliferation marker Proliferating cell nuclear antigen (Pcna[pos]) at 8dpf. We found no difference with WT larvae (Supplementary Fig. S3F, G). Finally, we quantified NSPC proliferation in the adult pallium. The proportion of Pcna[pos], Sox2[pos] cells in TgKI[n3-AG/n3-AG] homozygotes was ∼10–15%,

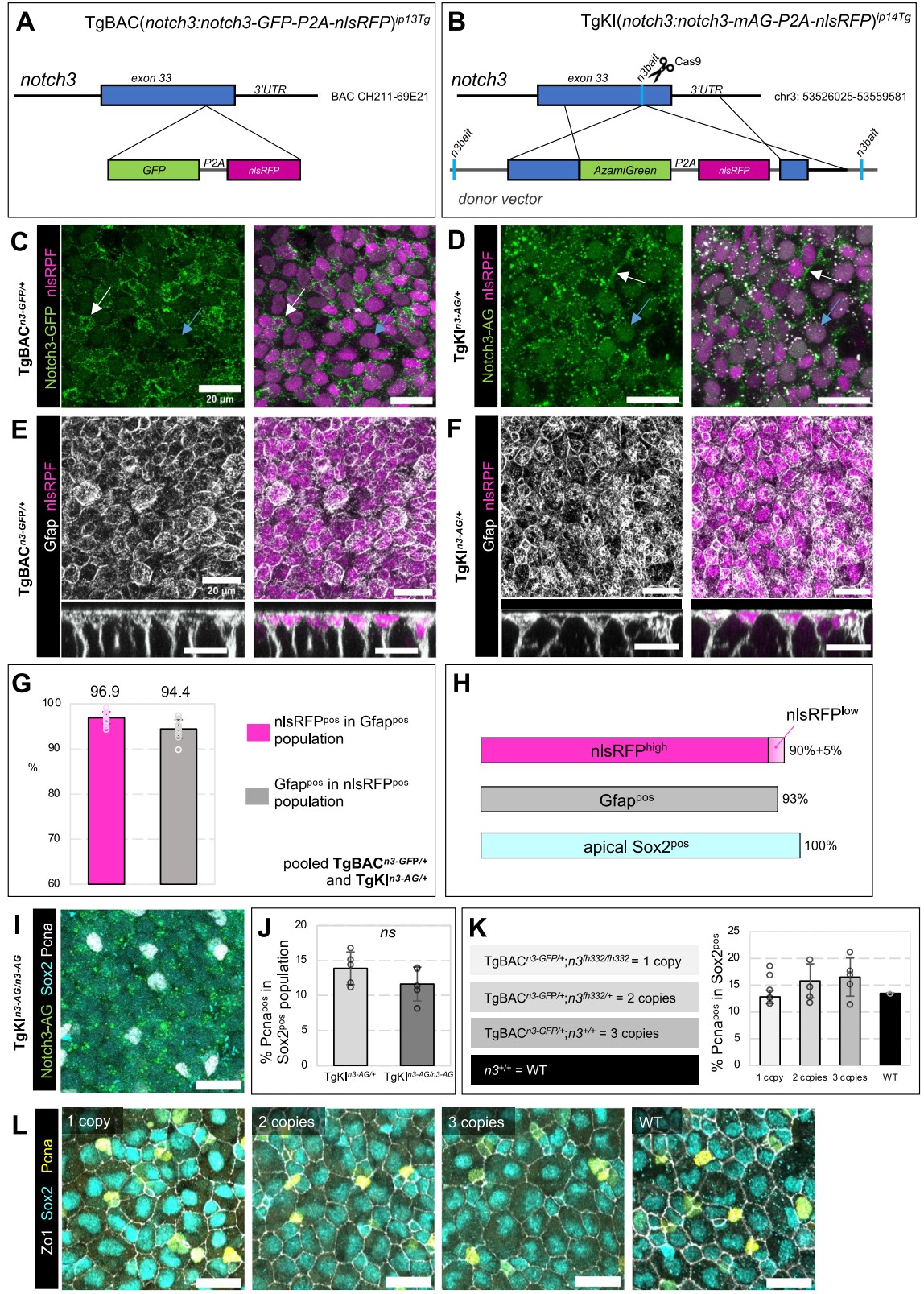

identical to heterozygotes and to previous reports in WT, and in agreement with the long quiescence phases of adult NSCs (Fig. 1I, J). Thus, Notch3-AG appears functional in its control of forebrain neurogenesis.

To assess Notch3-GFP, we crossed *notch3^{fh332}* mutants and TgBAC^{n3-GFP/+} fish to construct genetic contexts where *n3-GFP* is the only productive *notch3* copy (TgBAC^{n3-GFP/+};*notch3^{fh332/fh332}*) or is

combined with one WT *notch3* copy (TgBAC^{n3-GFP/+};*notch3^{fh332/+}*). Notch3-GFP alone was sufficient to rescue the curved larval phenotype at 7dpf (Supplementary Fig. S3I), and viability until adulthood (Supplementary Fig. S3J), thus behaved similarly to a single productive copy of endogenous *notch3* in *notch3^{fh332/+}* heterozygotes. We also used TgBAC^{n3-GFP/+}, carrying three copies of *notch3*, to assess the effect of Notch3-GFP overexpression on telencephalic neurogenesis. RG

**Fig. 1 | Notch3 fusion alleles are functional in adult NSCs. A, B** Transgenesis methods. **A** Recombination of *GFP-P2A-nlsRFP* in frame with *exon 33* of *notch3* into *BAC CH211-69E21*, followed by transposase-mediated recombination. **B** CRISPR/Cas9-mediated knock-in into endogenous *notch3*. A gRNA targeting the last *notch3* exon ("n3bait") was used to knock-in *AG-P2A-nlsRFP* in frame into *exon 33* by Homology Directed Repair. **C, D** Confocal dorsal views of dissected telencephala from 4 months post-fertilization (mpf) TgBAC[n3-GFP/+] and TgKI[n3-AG/+] fish stained by whole-mount immunohistochemistry (IHC) for GFP, AG or nlsRFP (color-coded). Cells expressing *notch3* show nuclear RFP (magenta). Notch3 protein appears on membranes (white arrows) and into some nuclei (blue arrows). **E, F** Similar views stained for Gfap and nlsRFP (color-coded; bottom: cross sections). **G** Quantification of cells positive for Gfap and/or nlsRFP in TgBAC[n3-GFP/+] and TgKI[n3-AG/+] fish (TgBAC[n3-GFP/+]: *n* = 1666 cells, 7 hemispheres from 4 fish; TgKI[n3-AG/+]: *n* = 974 cells, 5 hemispheres from 3 fish). **H** Proportion of Gfap[pos] (Zrf1 antibody, gray) and nlsRFP[pos] (magenta, high and low nlsRFP levels) cells among apical Sox2[pos] cells (cyan). **I** Confocal whole-mount dorsal view of an adult TgKI[n3-AG/n3-AG] telencephalon with IHC for Sox2, AG and Pcna (color-coded). **J** Percentage of Pcna[pos] cells among Sox2[pos] cells in adult TgKI[n3-AG/+] heterozygotes vs TgKI[n3-AG/n3-AG] homozygotes. ns: non-significant (Wilcoxon test, *p*-value = 0.16) (TgKI[n3-AG/+]: *n* = 1781 cells, 5 hemispheres from 4 fish; TgKI[n3-AG/n3-AG]: *n* = 1595 cells, 5 hemispheres from 4 fish). **K, L** Crossing TgBAC[n3-GFP/+] into the *notch3[fh332]* background generated genotypes carrying *notch3-GFP* and in total 1, 2 or 3 functional copies of *notch3* (color-coded, shades of gray) compared with WT (black). **K** Percentages of Pcna[pos] cells among Sox2[pos] (ANOVA test, *p*-value = 0.32, ns between genotypes) (1 copy: *n* = 760 cells, 2 fish; 2 copies: *n* = 1507 cells, 4 fish; 3 copies: *n* = 1243 cells, 3 fish; WT: *n* = 558 cells, 1 fish). **L** Confocal whole-mount dorsal views of adult telencephala of the different genotypes with IHC for Sox2, Zo1 and Pcna (color-coded). Scale bars: 20 μm. Panels 1C, D are representative of > 10 independent experiments; panels 1E, F are representative of 2 independent experiments. Bar plots (**G, J, K**) show mean values +/− SD, with individual hemisphere values as dots. All source data are provided as a Source Data file.

---

proliferation and neuronal generation were normal in TgBAC[n3-GFP/+] embryos (Supplementary Fig. S3B, D, E), but there was a strong trend for RG activation decrease in 8dpf larvae (Supplementary Fig. S3F, H). Thus, increased *notch3* gene dosage at this stage likely increases signaling and Notch3-GFP promotes quiescence entry, like WT Notch3. Finally, in the adult pallium, we quantified Pcna[pos] cells among all NSPCs (Sox2[pos];Zo1[pos]) in these different genotypes. There were no biases compared to WT (Fig. 1K, L). We conclude that Notch3-GFP is functional. The lack of a visible effect of *notch3* overexpression in the adult pallium of TgBAC[n3-GFP/+];*notch3[+/+]* fish will be discussed below.

### Notch3 fusion lines are dynamic and quantitative readouts of Notch3 signaling in adult pallial NSPCs

We next assessed whether the TgKI[n3-AG/+] and TgGFP[n3-GFP/+] lines could reliably quantify canonical Notch3 signaling in adult pallial NSPCs. Depending on signaling strength, we expect to observe NSPCs with various levels of nuclear N3ICD-fluo (Fig. 2A, B). At the population scale in TgKI[n3-AG/+] and TgBAC[n3-GFP/+] fish, the AG or GFP fluorescence signals at the cell membrane appeared much higher than the nuclear signal, as expected due to the high instability of N3ICD (Fig. 1C, D). We thus used nlsRFP, encoded in *notch3*-expressing cells, to develop a pipeline of nuclear segmentation to selectively quantify nuclear N3ICD-AG or N3ICD-GFP fluorescence (Fig. 2C). In both lines, nuclear fluorescence levels, measured as pixel intensity levels in individual cells across the NSPC population, displayed a broad range of values with an approximately normal distribution. It was similar in direct fluorescence and when using whole-mount IHC with anti-AG or anti-GFP antibodies (Fig. 2D, E), validating IHC for quantitative measurements.

We next tested whether N3ICD-fluo levels were quantitative readouts of Notch3 signaling. Because both lines affect predicted PEST domains of N3ICD, which normally facilitate Notch intracellular domain degradation by the ubiquitin ligase Fbw7, we used a panel of assays to assess their quantitative value. First, nuclear N3ICD-AG levels appeared on average two-fold higher in TgKI[n3-AG/n3-AG] homozygotes than in TgKI[n3-AG/+] heterozygotes (Fig. 2F). Second, nuclear N3ICD-GFP levels were significantly lower in proliferating (Pcna[pos]) than in quiescent (Pcna[neg]) cells, in agreement with Notch3 signaling promoting quiescence (Fig. 2G, left panel). In this experiment, measuring GFP levels in non-apical Sox2[pos], nlsRFP[neg] or nlsRFP[low] cells was also used to estimate the "zero" value of nuclear N3ICD-GFP, corresponding to IHC and imaging noise (Fig. 2G, right panel). By comparison, this further indicates that Pcna[pos] cells have N3ICD-GFP values still above null values, thus do experience N3ICD signaling, albeit at a low level (Fig. 2G). Finally, decreasing canonical Notch signaling for 24 hours using the gamma-secretase inhibitor LY411575 in vivo led to an accumulation of N3ICD-GFP at the membrane and a significant reduction of nuclear N3ICD-GFP levels (Fig. 2H, I). The range of decrease (1.7-fold from mean to mean at the population level), compared to the zero

value (Fig. 2G, right panel), further suggests that Notch3 signaling is lowered but not fully blocked by such a treatment.

Together, these results demonstrate that TgBAC[n3-GFP/+] and TgKI[n3-AG/+] permit a quantitative and direct readout of transcriptional Notch3 signaling in adult NSPCs in situ.

### Notch3 signaling levels in adult pallial NSPCs are controlled by ligand availability in addition to gene dosage

We took advantage of the quantitative power of our measures to assess the link between *notch3* gene dosage and signaling levels in adult pallial NSPCs. For this, we quantified nuclear N3ICD-GFP levels in TgBAC[n3-GFP/+] animals carrying one, two or three gene copies of *notch3* producing a functional protein (as in Fig. 1K). In the case of non-limiting amounts of ligands and signaling levels determined by gene dosage, we expect the same absolute amounts of nuclear N3ICD-GFP for these three genotypes (Fig. 2J, top); in contrast, if endogenous ligand availability is a limiting factor for signaling, the absolute levels of N3ICD-GFP should progressively decrease as *notch3* gene copy number increases (Fig. 2J, bottom). In support of the latter hypothesis, we observed progressively decreasing N3ICD-GFP levels between TgBAC[n3-GFP/+];*notch3[fh332/fh332]*, TgBAC[n3-GFP/+];*notch3[fh33/+]*; and TgBAC[n3-GFP/+];*notch3[+/+]* compound adults (Fig. 2K). Thus, ligand availability is a major determinant of N3ICD signaling levels in adult NSPCs, limiting the effect of the extra gene dose in the TgBAC[n3-GFP/+];*notch3[+/+]* genotype.

### Notch3 signaling levels are spatially patterned among adult NSPCs in situ

To gain insight into Notch3 signaling regulation, we used spatial statistics to describe the profile of N3ICD levels in space in adult TgKI[n3-AG/+] and TgBAC[n3-GFP/+] pallia. Our previous work showed that focusing on the cell neighborhood in the apical plane of NSPCs was sufficient to read out meaningful interactions controlling the pattern of activation events in physical space[27]. We observed no major variations in the z position of nuclei, and assimilated cell centers to the position of nuclei projected onto the apical plane. Then, with a marked L-function, we scored Notch3 signaling levels in the respective neighborhood of cells with highest vs lowest N3ICD values (highest or lowest 20% quantile of the whole distribution) and compared them with levels expected assuming uncorrelation, i.e., upon random cell permutations (Fig. 3A–C and Supplementary Fig. S4C). Because such analyses are based on distances between cell nuclei, we also considered apical cell size, measured using Zo1. At the population scale, cells of lowest Notch3 signaling levels tended to have smaller apical areas (and conversely, cells of high Notch3 signaling levels tended to be large) (Supplementary Fig. S4A, B). Nevertheless, there was no size bias within a 30 μm distance from N3ICD[low] cells (Fig. 3D, bottom panel) and, within the first 7-8 μm surrounding each N3ICD[low] cell, N3ICD

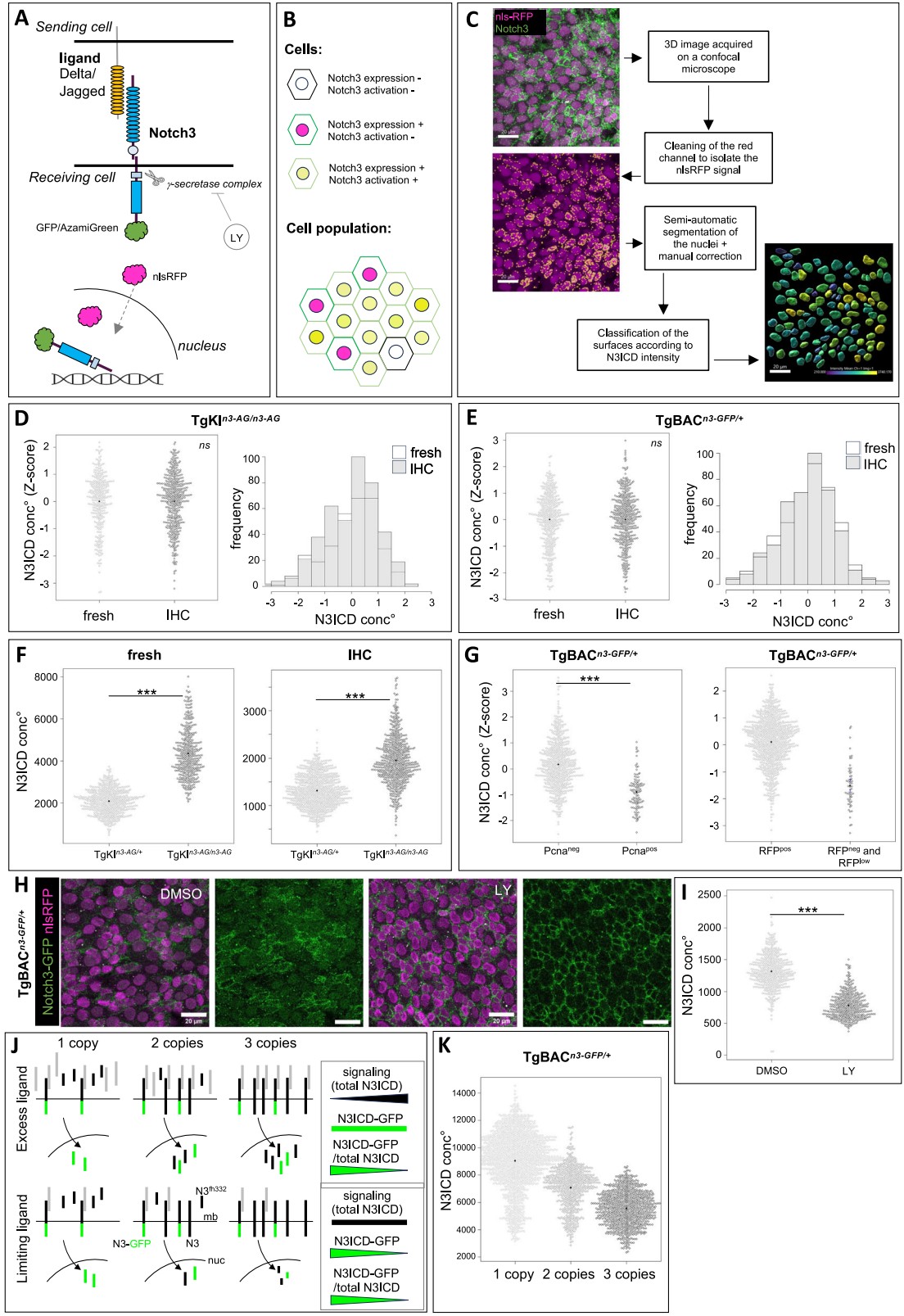

signaling levels (normalized to nuclear volume) were overall higher than expected by chance (Fig. 3D and Supplementary Fig. S4D, top panels). The trend was similar when considering absolute N3ICD values (within the hypothesis rejection threshold but significant at 5%, $p = 0.03$) (Fig. 3D and Supplementary Fig. S4D, middle panels). Thus, Notch3 signaling levels in individual cells across the germinal pallial population are spatially regulated: cells receiving the lowest

signaling tend to be surrounded by cells with the highest signaling.

## Jagged and Delta are differentially expressed in relation with NSPC state

To exploit quantitative and spatial information and mechanistically dissect the regulation of Notch3 signaling, we used the most

**Fig. 2 | Notch3 fusion alleles are dynamic and quantitative reporters of Notch3 nuclear signaling in adult NSPCs. A** Schematic: in TgBAC$^{n3\text{-}GFP/+}$ and TgKI$^{n3\text{-}AG/+}$, nlsRFP is separately translated and translocated into the nucleus. Notch3-fluo is membrane-bound: after ligand interaction, it is cleaved by γ-secretase and translocated into the nucleus (Notch3 intracellular domain - N3ICD-Fluo). γ-secretase is inhibited by LY-411575 (LY). **B** Expected Notch3-fluo pattern: *notch3* expressing cells have a red nucleus (nlsRFP, magenta) and a green plasma membrane (membrane-bound Notch3-fluo). Activation produces a yellow nucleus (green and red). **C** Segmentation method to quantify nuclear N3ICD-fluo using the Imaris software (Oxford Instruments). A mask removes strong red artefacts before 3D nuclear segmentation and manual correction. N3ICD-fluo quantification is typically performed on 100 to 200 nuclei per hemisphere, in the medial (Dm) region of the pallium (right: nuclei color-coded according to green intensity). Representative images from more than 10 independent experiments. **D, E** N3ICD-fluo concentration (conc°, total pixels intensity normalized to nuclear volume) in pallial NSPCs in adult TgKI$^{n3\text{-}AG/n3\text{-}AG}$ (**D**) and TgBAC$^{n3\text{-}GFP/+}$ (**E**), without ("fresh") or after ("IHC") antibody staining for AG or GFP, respectively (TgKI$^{n3\text{-}AG/n3\text{-}AG}$: $n = 317$ cells for "fresh" and $n = 428$ cells for "IHC"; TgBAC$^{n3\text{-}GFP/+}$: $n = 422$ cells for "fresh" and $n = 431$ cells for "IHC"; 3 brains / condition / genotype). Beeswarm plots show individual nuclei. Histograms: frequency distribution of N3ICD conc° intensities among all *notch3*$^{pos}$ cells. X test on frequencies, ns, *p*-value = 0.95 for TgKI$^{n3\text{-}AG/n3\text{-}AG}$ and 0.65 for TgBAC$^{n3\text{-}GFP/+}$. **F** N3ICD-AG conc° in TgKI$^{n3\text{-}AG/+}$ vs TgKI$^{n3\text{-}AG/n3\text{-}AG}$ in pallial NSPCs imaged using the same confocal parameters without or after IHC. Wilcoxon test, *p*-value < 2.2e-16 (TgKI$^{n3\text{-}AG/+}$: $n = 368$ cells « fresh », $n = 571$ cells « IHC »; TgKI$^{n3\text{-}AG/n3\text{-}AG}$: $n = 501$ cells « fresh », $n = 725$ cells « IHC »; $n = 3$ and 4 brains / genotype, respectively). **G** Left: Nuclear N3ICD-GFP conc° in non-proliferative (Pcna$^{neg}$, $n = 565$ cells) vs proliferative (Pcna$^{pos}$, $n = 103$ cells) pallial NSPCs in TgBAC$^{n3\text{-}GFP/+}$ ($n = 2$ brains). Wilcoxon test, *p*-value < 2.2e-16. Right: N3ICD-GFP conc° (Z-score values) in "RFP$^{neg}$ and RFP$^{low}$" cells ($n = 55$ cells), compared to values in RFP$^{pos}$ cells ($n = 782$ cells), allow to set a range of intensities for background noise (from 1 to 3 standard deviations under the mean) ($n = 3$ brains from TgBAC$^{n3\text{-}GFP/+}$ fish, segmentation on Sox2$^{pos}$ nuclei). **H, I** TgBAC$^{n3\text{-}GFP/+}$ fish were treated with 10 mM LY-411575 (LY) or carrier (DMSO) for 24 h. **H** Whole-mount brains (3 per condition) imaged « fresh » using the same confocal parameters (scale bars: 20 μm). Membrane accumulation of GFP signal at the membrane can be seen in LY samples. **I** Quantification of nuclear N3ICD-GFP conc° (segmentation on nlsRFP signal) (DMSO: $n = 474$ cells, LY: $n = 383$ cells). Wilcoxon test, *p*-value < 2.2e-16. **J, K** Evaluation of ligand dose-dependency in Notch3 signaling. TgBAC$^{n3\text{-}GFP/+}$ fish were combined with *notch3*$^{fh332}$ mutants to generate TgBAC$^{n3\text{-}GFP/+}$ fish with 1, 2 or 3 functional *notch3* copies. **J.** Theoretical predictions on the amount of nuclear N3ICD-GFP for each genotype. Bottom: limiting ligands: signaling (total amount of N3ICD, tagged or not) is constant while the absolute nuclear level of N3ICD-GFP, and its ratio over total nuclear N3ICD (N3ICD-GFP/[N3ICD + N3ICD-GFP]), decrease as *notch3* copies increase (bar codes on the right). Top: excess ligands: signaling increases as *notch3* copies increase, while nuclear N3ICD-GFP levels remain constant, and the ratio of N3ICD-GFP over total nuclear N3ICD decreases. Ligands, Notch3 (N3), Notch3-GFP (N3-GFP) and Notch3$^{fh332}$ (N3$^{fh332}$) proteins are color-coded; mb: cell plasma membrane, nuc: nucleus. **K** Nuclear N3ICD-GFP levels ("N3ICD-GFP" in (**J**), measured as conc°, i.e., normalized over nuclear volume) in the three genotypes (1 copy: $n = 1344$ cells, 4 brains; 2 copies: $n = 544$ cells, 3 brains; 3 copies: $n = 751$ cells, 3 brains; segmentation performed on Sox2 expression) (Welch's ANOVA test, *p*-value < 2.2e-16). All source data are provided as a Source Data file.

---

appropriate of the two transgenic lines based on technical needs: TgBAC$^{n3\text{-}GFP/+}$ NSPCs are less crowded with trafficking vesicles, facilitating quantifications, while TgKI$^{n3\text{-}AG/+}$ can be crossed into GFP reporter lines.

First, we analyzed the expression of Notch ligands. The zebrafish genome contains four *delta* (*dl*) genes (*dla*, *dlb*, *dlc* and *dld*) and three *jagged* genes (*jag1a*, *jag1b* and *jag2b*), and scRNAseq[25] predicts detectable expression of *dla* and *jag1b* in adult pallial NSCs (Supplementary Fig. S5A–G). We validated these findings using sensitive single-molecule fluorescent ISH (smFISH) with RNAscope probes on whole-mount adult pallia. This highlighted *dla* and *jag1b* in NSPCs, while *jag1a* was not expressed and *jag2b* was restricted to neurons, below the NSPC layer (Supplementary Fig. S5H), confirming and extending previous work[23].

Strikingly, the expression of *dla* and *jag1b* appeared different both in intensity and distribution. *dla* is expressed with a broad range of intensities, measured by the number of RNAscope dots (interpreted to reveal individual RNA molecules), with an exponential decrease from null/low levels to very high levels (Fig. 4A, B). In situ, *dla*$^{high}$ cells displayed a salt-and-peppery distribution similar to the pattern previously described in Tg(*dla:gfp*) fish (Fig. 4A and Supplementary Fig. S5A)[27,44]. In contrast, *jag1b* expression levels followed a normal distribution over a narrower range of levels, with most NSPCs expressing some *jag1b* transcripts (Fig. 4A, B and Supplementary Fig. S5C). Overall, there was no correlation between *dla* and *jag1b* expression levels at the individual cell level (Fig. 4C). Like for GFP in the adult pallium of Tg(*dla:gfp*) fish[44], we confirmed significantly higher *dla* transcript levels in Pcna$^{pos}$ compared to Pcna$^{neg}$ cells (Fig. 4D). The relationship between *jag1b* expression levels and cell proliferation was opposite, quiescent cells expressing significantly higher levels of *jag1b* (Fig. 4E).

**Nuclear N3ICD levels cell-autonomously correlate with *jagged1b* but not *deltaA* expression, and sign NSPC position along lineage and quiescence progression**

To elaborate hypotheses on Notch3 signaling regulation, we directly combined smFISH for *jag1b* or *dla* with N3ICD quantifications in TgBAC$^{n3\text{-}GFP/+}$. We found a clear positive correlation between *jag1b* expression and nuclear N3ICD levels in individual cells (Fig. 4F, G), while there was no correlation between *dla* and N3ICD (in fact, most cells with

high *dla* expression are N3ICD$^{low}$) (Fig. 4H, I). For comparison with previous work[27,44], we also exploited TgKI$^{n3\text{-}AG}$ crossed into Tg(*dla:gfp*)[45] to quantify N3ICD-AG vs GFP (Fig. 4J). We confirmed that *dla*:GFP$^{neg}$;nlsRFP$^{pos}$ cells on average display significantly higher N3ICD levels than *dla*:GFP$^{pos}$;nlsRFP$^{pos}$ cells (Fig. 4K). This also correlates with the generally larger apical size of *dla*:GFP$^{neg}$ compared to *dla*:GFP$^{pos}$ cells mentioned above (Supplementary Fig. S4A, B), and with our previous observations that apical size decreases with lineage progression[44].

Among NSCs, *dla*:GFP expression differentiates between subpopulations endowed with self-renewal (*dla*:GFP$^{neg}$) versus neurogenic (*dla*:GFP$^{pos}$) potential, the latter showing increased GFP levels as they progress along the lineage towards neurogenesis commitment[44]. We found that, within *dla*:GFP$^{pos}$; nlsRFP$^{pos}$ cells, N3ICD levels decreased as *dla*:GFP levels increased (Fig. 4L). Within each *dla*:GFP$^{pos}$ and *dla*:GFP$^{neg}$ subpopulation, our previous observations based on direct cell tracking by intravital imaging showed that NSCs with larger apical area are statistically closer to their next division[44]. Within each of these subpopulations in nlsRFP$^{pos}$ cells, N3ICD levels generally increased with apical area size -but stopped increasing linearly as apical areas reached high values- (Fig. 4M, N).

Together, these results also connect N3ICD levels with ligand expression and the position of individual NSPCs along their lineage and quiescence progression (Supplementary Fig. S5I).

**Experimental downregulation of Notch signaling highlights paradoxical responses of *jag1b*/*hey1* vs *dla* expression**

Notch3 signaling in NSCs was proposed to involve lateral inhibition originating from Dla$^{pos}$ NP neighbors[27]. Modeling lateral inhibition in a cell lattice also suggested that signaling intensity depends on the length of Notch/Delta-contacting membranes[46]. To test whether such interactions could suffice to explain the quantitative N3ICD pattern, we quantified nuclear N3ICD-AG in TgKI$^{n3\text{-}AG/+}$;Tg(*dla:gfp*) for each NSPC relative to its number of *dla:gfp*$^{pos}$ direct neighbors (i.e., sharing a membrane segment at their apicobasal interface, revealed with Zo1 IHC)[27] or its total number of neighbors. We found that there was no correlation between N3ICD levels and these parameters (Supplementary Fig. S6A, B). Thus, the Notch3 activity pattern in adult pallial NSPCs likely involves more than a simple lateral inhibition model predicted by the length of Dla$^{pos}$ cell contacts.

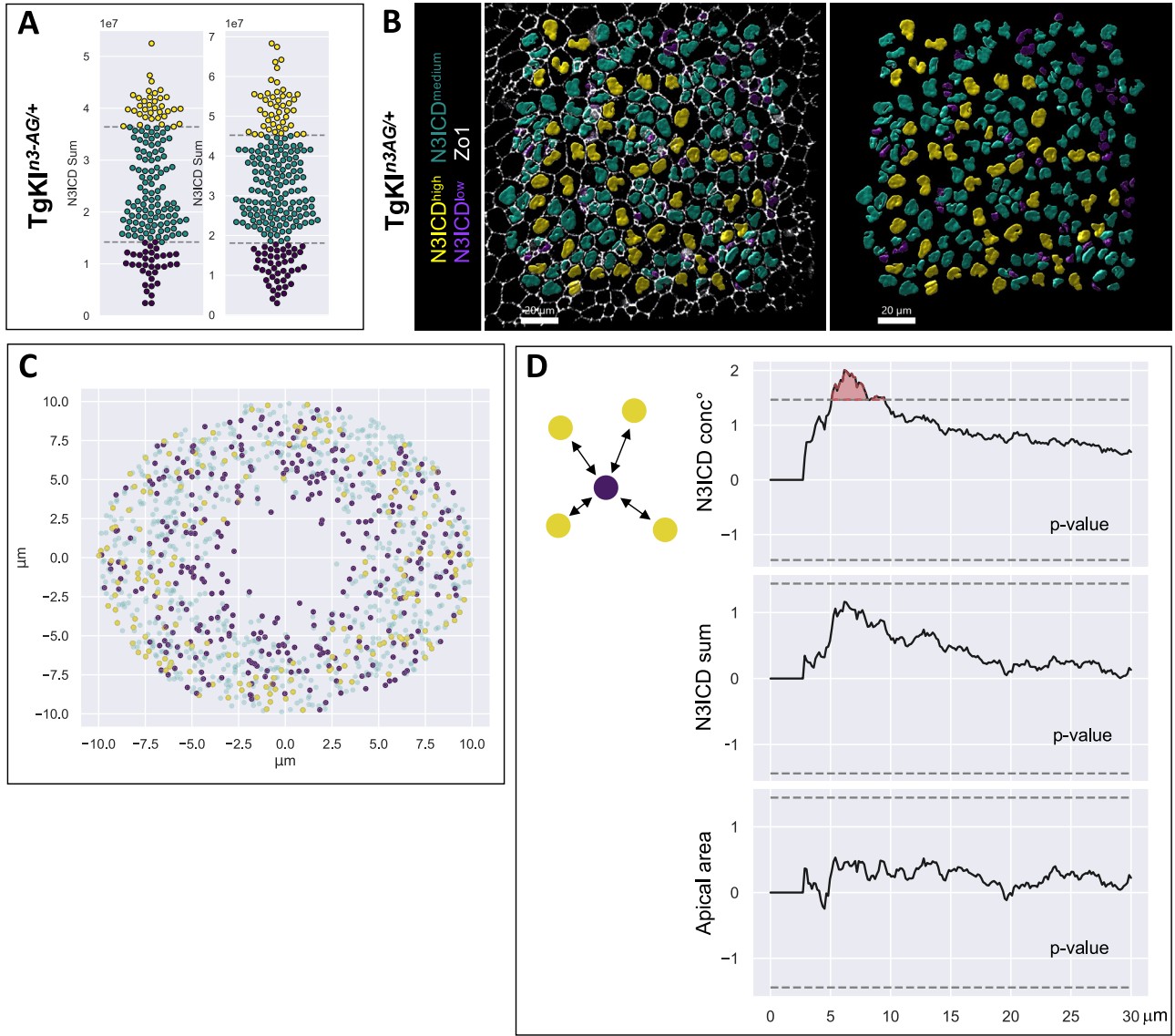

**Fig. 3 | N3ICD levels follow a non-random spatial distribution pattern across the NSPC population. A**, **B** TgKI[n3-AG/+] adult pallia were processed for IHC for AG, Zo1 and Sox2 to classify cells according to their nuclear N3ICD-AG levels. **A** Beeswarm plot showing the distribution of N3ICD-AG levels in two pallial hemispheres and its subdivision into cells of high (top 20% quantile, yellow), low (bottom 20% quantile, purple) and intermediate (green) Notch3 signaling. **B** Same color code shown on the corresponding in situ images showing Sox2[pos] segmented nuclei (left: with Zo1; right: without). Scale bars: 20 μm. Representative image of more than 10 independent experiments. **C** Superposed neighborhoods of all N3ICD[low] cells. For each cell in this group, all the cells surrounding it at a maximal radius of 10 μm are plotted at their position with respect to the original cell, color-coded as in (**A**, **B**). **D** Deviation from marked L-functions from an uncorrelated model as a function of the distance (radius in micrometers, x axis) from cells of low N3ICD values (scheme on left). Identity of the mark from top to bottom: N3ICD conc° (i.e., sum normalized to nuclear volume) per nucleus, N3ICD sum (absolute value) per nucleus, apical area. n = 401 cells from 2 hemispheres (173 + 228 cells) from 1 brain. Dashed lines: hypothesis rejection threshold, with significant deviations in red. *p*-values: N3ICD conc° = 0.01; N3ICD sum = 0.06; apical area = 0.89.

The similar distribution of Notch3 activity levels and *jag1b* expression is suggestive of lateral induction. Reinforcing this possibility, we identified several RBPj binding sites in the second exon of *jag1b*, including 2 in a head-to-head configuration reported to efficiently bind RBPj[47] (Supplementary Fig. S6C). To test this hypothesis, we first assessed in situ the response of *dla* and *jag1b* to lowering Notch signaling by gamma-secretase inhibition. In the case of lateral inhibition, lowering Notch should increase ligand transcription; conversely, for lateral induction, decreased ligand transcription is expected (Supplementary Fig. S6D). Although LY treatment is not selective of Notch3 but will potentially affect signaling by other Notch receptors, Notch3 is the predominant Notch receptor in quiescent adult NSCs (Supplementary Fig. S6E, and ref. 24). We treated TgBAC[n3-GFP/+]

adults with 10 μM LY511475 for 24 h and assessed *dla* and *jag1b* expression using smFISH together with Zo1 revealed by IHC (Fig. 5A). This treatment decreases N3ICD-GFP to almost half (as in Fig. 2I), but NSCs do not yet enter the cell cycle[48] (Fig. 5B), avoiding confounding measurements. We observed that the average levels of *dla* increased, while expression levels of *jag1b* dropped dramatically (Fig. 5C, D). These observations support Dla vs Jag1b being respectively engaged in lateral inhibition vs lateral induction with Notch.

## Jagged sustains Notch3 signaling and the progenitor state in adult NSCs

Next, to assess the role of Jag1b in Notch3 signaling in adult pallial NSCs, we tested whether Jag1b knockdown affects N3ICD levels. We

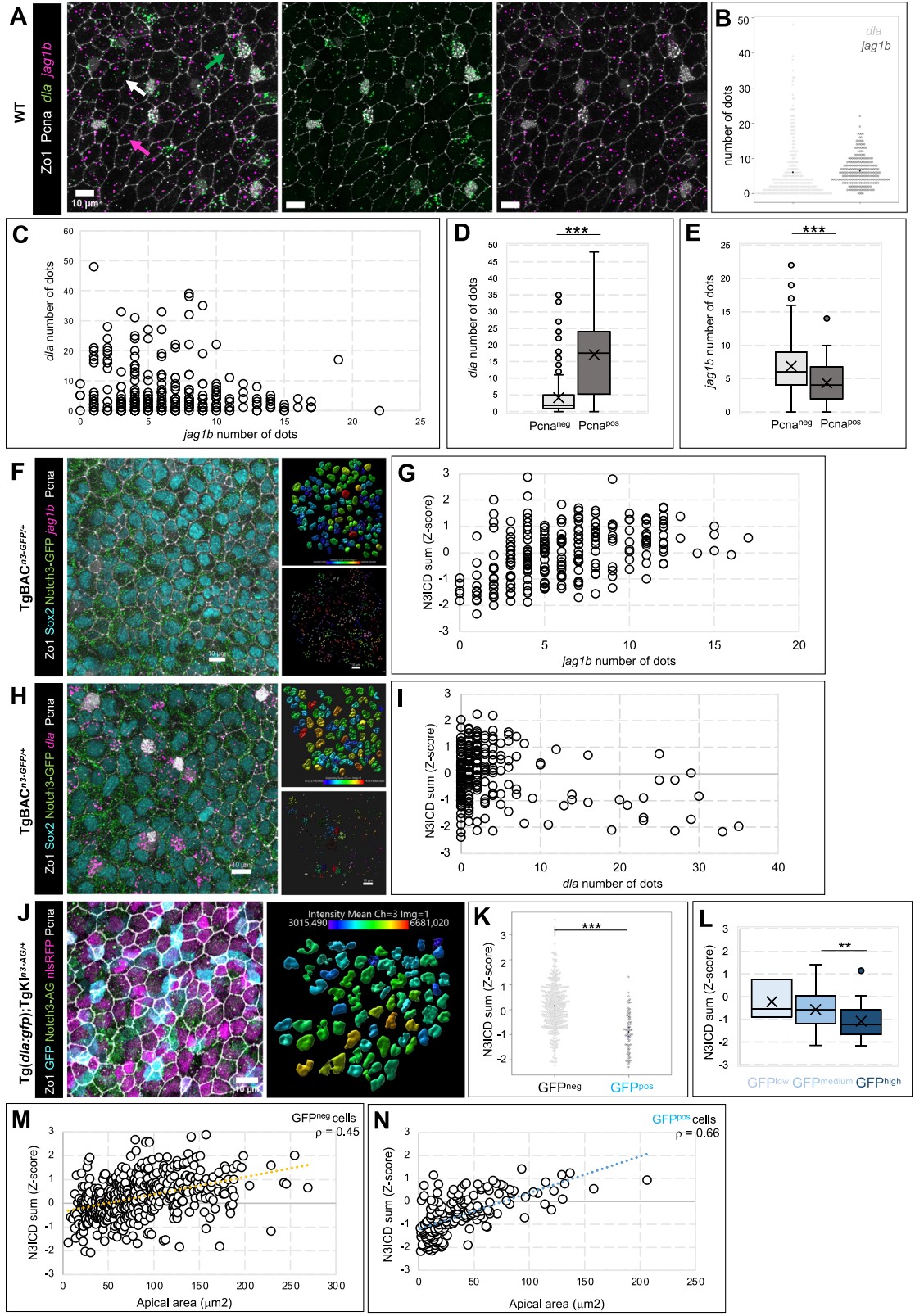

used first a lissamine-tagged version of the *jag1b* morpholino (MO) validated in previous studies[49], using electroporation to transduce NSCs in vivo[50]. However, this approach reduces *jag1b* in isolated cells, and phenotypes might be confounded by the complex feedback interactions on N3ICD due to the presence of several cell neighbors and ligands. We therefore used the cell-permeable "vivo-MO" version of *jag1b*MO, injected intracranially into the brain

ventricle of TgKI*n3-AG/+* adults, to target all cells with ventricular contact[51,52] (Fig. 5E). At 1-day post-injection (dpi), we quantified nuclear N3ICD-AG levels in Sox2*pos* cells (Fig. 5F), comparing *jag1b*MO to a standard control vivo-MO (*ctrl*MO). N3ICD-AG was significantly reduced upon *jag1b* knockdown (Fig. 5F, G), indicating that Jag1b is necessary for maintaining global Notch3 signaling in adult pallial NSCs.

**Fig. 4 | *jag1b* and *dla* are differentially expressed in relation with N3ICD levels along quiescence and lineage progression. A** Expression of *dla* and *jag1b* (RNA-scope, green and magenta dots, respectively) on a whole-mount WT adult brain, with IHC for Zo1 and Pcna (white). Some cells express only one ligand (green and magenta arrows), others show co-expression (white arrow). **B, C** Quantification of ligand transcripts (dots counts) per cell of (A) (*n* = 268 cells, 1 brain; Spearman's ρ = - 0.12). **D, E.** Quantification of *dla* (Wilcoxon test, *p*-value < 2.2e-16) and *jag1b* (Wilcoxon test, *p*-value = 0.0004) transcripts in proliferating (Pcna^pos^, *n* = 229 cells) versus non-proliferating (Pcna^neg^, *n* = 40 cells) cells of (A). **F–I** RNAscope for *jag1b* (**F**) or *dla* (**H**) on whole-mount TgBAC^n3-GFP/+^ brains, with IHC for Notch3-GFP, Sox2, Zo1 and Pcna (color-coded, left). Nuclei are segmented on Sox2 staining (top right – color-code for N3ICD pixel sum intensity), and *jag1b* or *dla* dots are counted and manually assigned to each cell (bottom right, each color = one cell). *jag1b* expression and N3ICD intensity in individual cells (**G**) show a correlation that plateaus after 10 dots (2 brains, 234 cells; Spearman's ρ = 0.45); cells with high *dla* expression show low Notch3 activity (**I**) (2 brains, 246 cells; Spearman's ρ = 0.01). **J** *Tg(dla:gfp)*;TgKI^n3-AG/+^ brain with IHC for GFP, AG, Zo1, Pcna and nlsRFP (color-coded). Left: whole-mount confocal view; right: segmented nlsRFP^pos^ nuclei with N3ICD levels color-coded. Representative image from 2 independent experiments. **K** Corresponding quantification of N3ICD in GFP^pos^ vs GFP^neg^ cells (Z-score of N3ICD pixel sum; 3 brains; *n* = 416 and 67 cells, respectively). Wilcoxon test, *p*-value = 3.045e-15. **L** Quantification of N3ICD in *dla*^pos^ cells relative to GFP (low, medium or high, color-coded) (*n* = 3, 164 and 28 cells, respectively, 3 brains; Welch's ANOVA test on *dla*:GFP levels groups, *p*-value = 0.058; Wilcoxon test on medium vs high *dla*:GFP levels: *p*-value = 0.002). **M, N** Corresponding quantification of N3ICD in GFP^neg^ (**M**) vs GFP^pos^ (**N**) cells relative to apical area (Z-score of N3ICD pixel sum; 3 brains; *n* = 470 and 196 cells, respectively; Spearman's ρ = 0.45 and 0.66, respectively). All scale bars: 10 μm. Boxplots show median (central line), mean (X), interquartile range (box), whiskers (bars) and outliers (dots). All source data are provided as a Source Data file.

Notch3 signaling has two reported functions in adult pallial NSCs: the promotion of quiescence and the promotion of stemness[26]. We assessed the effects of *jag1b* knockdown on these activities using whole-mount IHC for readouts of cell activation (Pcna) and the progenitor state (Sox2) at 1dpi. Global proliferation levels (Pcna^pos^) remained unchanged (Fig. 5H). In contrast, the intensity of Sox2 expression was massively decreased -although ventricular cells remained Sox2^pos^ (Fig. 5I). There was also a small but statistically significant trend increase in *dla* transcription, more prominent in NSCs than in NPs (as assessed using apical size enrichment cut-offs)[44] (Fig. 5J, K). This *dla* phenotype differed in two respects with the increased *dla* transcription observed with LY treatment: it was quantitatively much reduced (Fig. 5D, K), and the *dla* salt-and-peppery expression pattern was preserved (Fig. 5J). These differences are striking, given that N3ICD levels are divided by approximately half in both cases.

Together, Jag1b contributes significantly to Notch3 signaling in situ, and Jag1b-mediated signaling impacts stemness but not the salt-and-peppery pattern of *dla* nor NSC proliferation rate.

### *hey1* expression responds to the Jag1b-Notch3 signaling axis in adult pallial NSCs

Effectors of canonical Notch signaling include Hairy/Enhancer-of-Split transcription factor genes. Several are expressed in pallial NSCs, among which *her4* genes and *hey1*, which are downregulated upon gamma-secretase inhibition at the adult stage, and in *notch3*^fh332/fh332^ larvae[25–27]. Of particular interest here was *hey1*, associated with Notch activation by Jagged in other contexts[53–57]. *hey1* proved expressed at variable levels in individual NSPCs (Supplementary Fig. S7A, B). To compare it with Jag1b/Notch3 signaling, we combined smFISH for *hey1* and *jag1b* in WT and TgBAC^n3-GFP/+^ fish. *hey1* expression levels showed a weak positive correlation with *jag1b* (Supplementary Fig. S7C, D) and with N3ICD levels (Supplementary Fig. S7E). Like *jag1b*, however, *hey1* was virtually expressed in all quiescent cells and was absent from proliferating cells (Supplementary Fig. S7F), and was dramatically reduced by LY treatment (Supplementary Fig. S7G, H).

The similar behavior of *hey1* and *jag1b* expression suggesting a functional relationship, we tested the effect of blocking *jag1b* on *hey1* expression, quantifying *hey1* expression in situ after intracranial injection of *jag1b*MO (Supplementary Fig. S7I). At 3dpi *hey1* expression was weakly decreased compared to fish treated with the *ctrl*MO (Supplementary Fig. S7J). These results position *hey1* directly or indirectly downstream of Jag1b/Notch signaling in adult pallial NSCs, but *hey1* expression clearly also responds to other inputs.

### Jagged-mediated Notch3 signaling potentiates lateral inhibition

The pattern of N3ICD signaling in physical space (Fig. 3D), and the observed coincidence of low N3ICD^low^ and *dla*:GFP^high^ cells (Fig. 4L), suggest that this spatial pattern results from (Dla-mediated) lateral inhibition. In addition, the Jag1b data presented above are compatible with Jag1b-mediated lateral induction. Previous modeling work predicts that, when lateral induction is superimposed to lateral inhibition, more robust pattern formation ensues where the Notch-Delta signaling difference between neighboring cells can become reinforced[58,59]. This outcome is dependent on the relative strength of lateral induction and lateral inhibition and is observed for intermediate to low lateral induction levels, when lateral inhibition dominates[58]. *jag1b*MO experiments, which decrease Jag1b at the population level, provide a context to address this prediction.

We first addressed whether the spatial pattern of N3ICD signaling levels was altered upon *jag1b* knockdown. We injected TgKI^n3-AG/+^ adults intracranially with the *ctrl*MO or the *jag1b*MO (Fig. 6A) and recorded N3ICD levels and the spatial location of each individual cell. We then applied the marked L-function as in Fig. 3D to measure the deviation of N3ICD levels compared to an uncorrelated pattern in the vicinity of N3ICD^low^ cells. *ctrl*MO-treated brains behaved as described in untreated fish (Fig. 6C), with N3ICD^low^ cells surrounded by higher N3ICD levels than expected. We found that this effect was not modified in *jag1b*MO brains (Fig. 6C). However, the distribution range of N3ICD levels was significantly decreased upon *jag1b*MO knockdown (variance under *ctrl*MO conditions: $5.0 \times 105$; under *jag1b*MO conditions: $2.5 \times 105$, *p*-value = $3.4 \times 10^{-16}$) (Fig. 5G). Thus, lowering Jag1b-mediated signaling preserves the structure of the Notch3 signaling pattern, but the difference between sender (N3ICD^low^) and receiver (N3ICD^high^) cells is decreased. These results are in fitting with predictions that superimpose lateral inhibition and lateral induction, and compatible with a role for Jag1b-mediated signaling in reinforcing lateral inhibition[58,59].

## Discussion

Together, we propose that N3ICD levels and their spatiotemporal distribution pattern among the NSPC population result from the differential but synergistic engagement of Notch3 with two ligands, together controlling different NSPC properties, their spatial distribution, and their temporal unfolding.

### Quantification of Notch3 signaling, allelic expression and dosage buffering in adult NSPCs in situ

TgBAC^n3-GFP/+^ and TgKI^n3-AG/+^ permit to selective readout Notch3 expression, subcellular localization and signaling in situ. In the absence of a reliable antibody, this is necessary to report the selective engagement of Notch3 vs Notch1 in NSPCs. Reporter lines based on RBPj binding elements do not distinguish between different Notch receptors. Further, major transcriptional changes occur in NSPCs within 24 h of Notch signaling blockade[48], stressing the importance of a fast and dynamic readout. An antibody against endogenous N3ICD would be needed to determine to which extent N3ICD-GFP and N3ICD-AG, which exclude one or both predicted PEST domains, mimic the

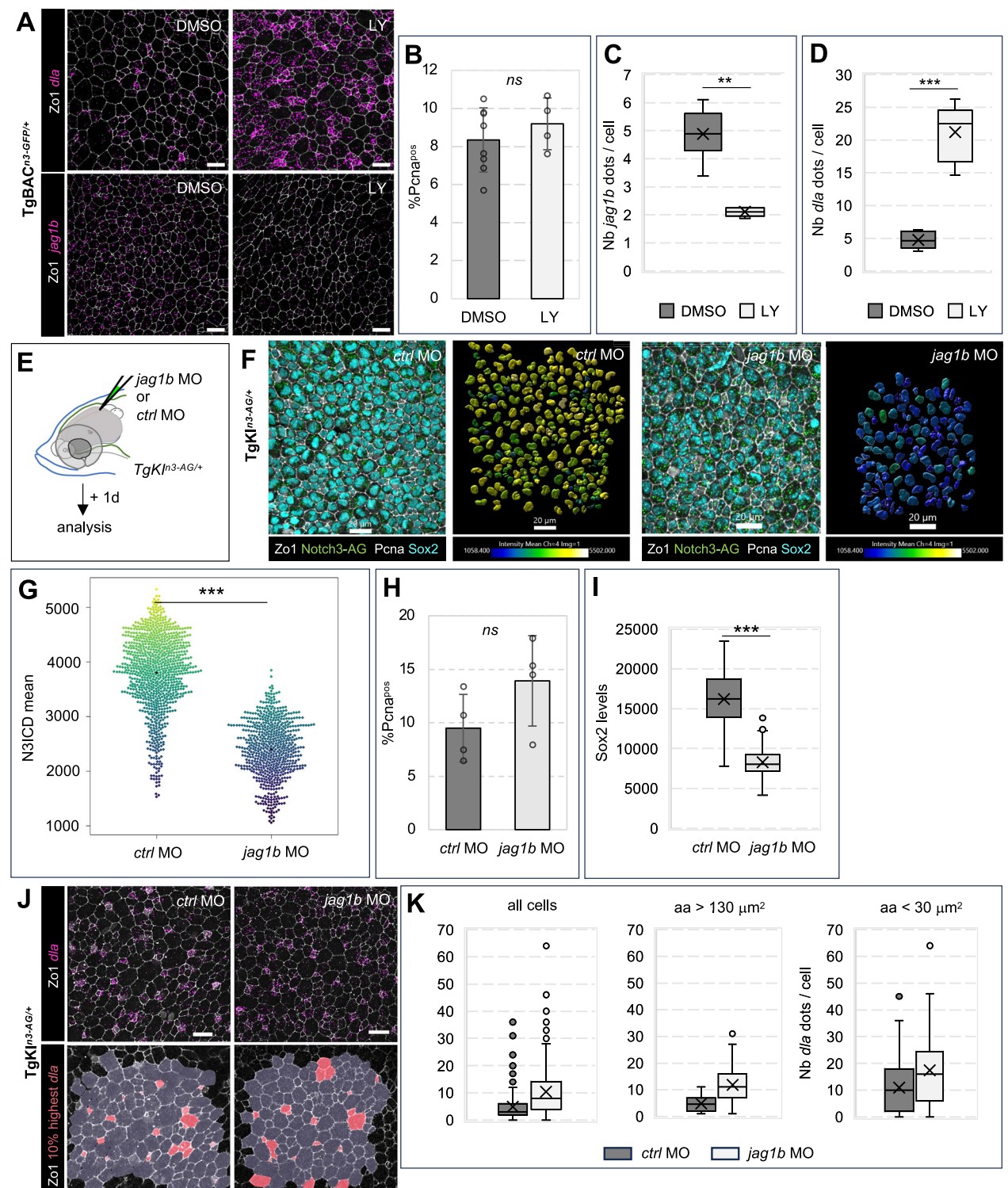

stability of endogenous N3ICD. In cocultures of Notch1- (or Notch2)- and Jagged-expressing mouse cells initiating Notch signaling, NICD half-life was reported to increase from 45 to 180 min in the absence of the cofactor Mastermind, which normally renders NICD sensitive to PEST-dependent degradation[60]. In adult pallial NSCs, increased stability of N3ICD in the range of some hours may slightly delay quiescence exit or the progression to downstream lineage states. This was not detectable within the limit of our measures, as our phenotypical analyses show that the functionality of both Notch3 fusion alleles is at least close to that of WT Notch3, suggesting largely preserved dynamics (Fig. 1 and Supplementary Figs. S2, S3).

Our work brings forward interesting observations comparing gene dosage, signaling levels and function in adult NSPCs. The fact that NSPCs in TgKI[n3-AG/+] adults overall harbor a half-dose of nuclear N3ICD-AG compared to TgKI[n3-AG/n3-AG] (Fig. 1J) indicates that both notch3 alleles are expressed equally to contribute to signaling. Tracking N3ICD-GFP in contexts expressing one, two or three functional gene copies of notch3 further supports that Notch3 signaling levels are buffered by ligand availability. This likely also explains why three functional gene copies do not lead to modified NSC properties in situ (Fig. 1K). Together, our findings stress the "dose /cell state" relationship of Notch3 signaling levels in adult NSCs, and its tight regulation by limiting ligands.

**Fig. 5 | Jag1b and Notch3 positively interact and maintain the progenitor state. A** Expression of *dla* (top) and *jag1b* (bottom) (RNAscope, magenta dots), with IHC for Zo1 and Pcna, Notch3-GFP and Sox2 (color-coded), in whole-mount TgBAC*n3-GFP/+* pallia after 24 h treatment with 10 mM LY-411575 or DMSO. **B–D** Quantifications in DMSO- vs LY-treated fish: percentage of Pcna*pos* cells (**B**) (DMSO: *n* = 2433 cells, 8 hemispheres from 4 brains; LY: *n* = 2438 cells, 4 hemispheres from 2 brains; Wilcoxon test, *p*-value = 0.38), *jag1b* expression (**C**) (DMSO: *n* = 2068 cells, 8 hemispheres from 4 brains; LY: *n* = 1762 cells, 5 hemispheres from 3 brains; Wilcoxon test, *p*-value = 0.002) and *dla* expression (**D**) (DMSO: *n* = 2580 cells, 8 hemispheres from 4 brains; LY: *n* = 1762 cells, 5 hemispheres from 3 brains; Wilcoxon test, *p*-value = 0.001). In (**C**, **D**), measurements are at population scale (total dots / number of apical surfaces). **E** TgKI*n3-AG/+* fish were injected with a "vivo morpholino" against *jag1b* (*jag1*MO) or a control *vivoMO* (*ctrl*MO) and analyzed after one day. **F** Whole-mount confocal views of pallia from *jag1*MO *jag1b*MO- or *ctrlMO*-injected fish immunostained for Notch3-AG, Sox2, Zo1 and Pcna (color-coded). Left: whole-mount views; right: nuclei segmented on Sox2 and color-coded for N3ICD-AG conc° (identical color scale in both conditions). **G** Distribution of N3ICD-AG conc° in each cell of *ctrl*MO- vs *jag1b*MO-injected brains (*n* = 876 and 637 cells, respectively, in 6 hemispheres from 3 brains for each condition) (Wilcoxon test, *p*-value < 2.2e-16). Color range as in (**F**). **H, I** Proportion of proliferating (Pcna*pos*) cells (**H** - *ctrl*MO: *n* = 670 cells, 4 hemispheres from 2 fish; *jag1b*MO: *n* = 535 cells, 4 hemispheres from 2 fish; Wilcoxon test, *p*-value = 0.11, ns) and Sox2 intensity (normalized to nuclear volume) (**I** - *ctrl*MO: *n* = 876 cells and *jag1b*MO: *n* = 637 cells, in 6 hemispheres from 3 brains for each condition; Wilcoxon test, *p*-value < 2.2e-16) among Sox2*pos* cells in *ctrl*MO- vs *jag1b*MO-injected brains. **J** Top: Whole-mount confocal views of pallia from *jag1b*MO- or *ctrl*MO-injected fish, showing *dla* expression (RNAscope, magenta dots) and Zo1 (IHC, white). Bottom: apical surfaces of the 10% cells with the highest *dla* expression (pink) (over a total of *n* = 280 and 278 cells for *ctrl*MO and *jag1b*MO, respectively). **K** *dla* dots per cell, in all cells (left) or in cells of large (middle) or small (right) apical surface area (aa) (60 and 71 cells of aa> 130 μm² and 57 and 45 cells of aa< 30 μm², respectively, in 2 hemispheres from 1 brain for each condition) (Wilcoxon tests: all cells *p*-value < 2.2e-16, aa> 130 μm² *p*-value = 7.51e-10, aa< 30 μm² *p*-value = 0.007). Scale bars: (**A**, **F**): 20 μm; (**J**): 10 μm. Bar plots (**B**, **H**) show mean values +/− SD with individual hemisphere values as dots. Boxplots show median (central line), mean (X), interquartile range (box), whiskers (bars) and outliers (dots). All source data are provided as a Source Data file.

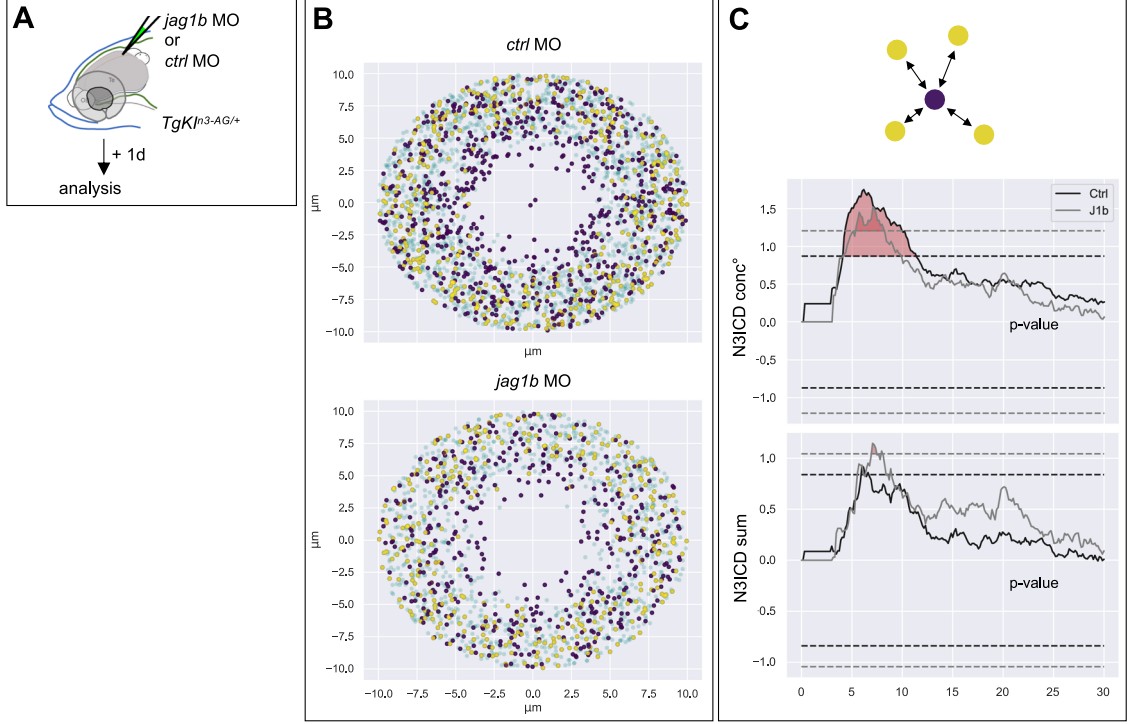

**Fig. 6 | Downregulation of Jag1b preserves the spatial structure of the Notch3 signaling pattern. A** TgKI*n3-AG/+* fish were injected with *jag1b*MO or *ctrl*MO. After 1 day, nuclear N3ICD-AG levels were quantified and cell position recorded. **B** Superposed neighborhoods of all N3ICD*low* cells, as in Fig. 3C. **C** Deviation from marked L-functions from an uncorrelated model as a function of the distance from cells of low N3ICD values (as in Fig. 3D, color-coded per condition: black: *ctrl*MO, gray: *jag1b*MO). Identity of the mark from top to bottom: N3ICD conc° per nucleus, N3ICD sum per nucleus. *ctrlMO*: *n* = 876 cells from 3 brains (6 hemispheres); *jag1b*MO: *n* = 256 cells from 1 brain (2 hemispheres). Dashed lines: hypothesis rejection thresholds (color-coded as above), with significant deviations in red. All *p*-values < 0.01.

## N3ICD levels correlate with NSC decisions and combine differential input from DeltaA and Jagged1b

Notch3 signaling levels vary from cell to cell along two NSC decision axes: (i) the quiescence/activation balance, and (ii) NSC lineage progression. Indeed, N3ICD levels (i) are significantly lower during activation (Fig. 2G), in agreement with Notch3 signaling promoting quiescence[24], and (ii) negatively correlate with *dlaA*:GFP levels (Fig. 4L), which globally read lineage progression[44]. These results support that N3ICD levels quantify "transcriptional signaling" in situ, in agreement with previous work in Drosophila and C. elegans where nuclear NICD levels quantitatively correlate with target genes activation[36–38].

*dla* and *jag1b* are the main ligands expressed within the pallial germinal population[23]. We reveal that their expression patterns strikingly differ, from each other and in their relationship to N3ICD levels: *dla* transcription is salt-and-peppery and *dla*[high] NSPCs are generally N3ICD[low], while *jag1b* is expressed broadly and positively correlates with N3ICD at the single cell level (*jag1b*[high] NSCs are N3ICD[high]) (Fig. 4). Although consolidation by analyses of the Dla and Jag1b proteins is needed, these patterns suggest different mechanisms of Notch3 signaling. The role of Dla-Notch3 signaling can be inferred combining previous work and this work: (i) at any given time, *dla*[pos] cells (some NSCs and all NPs) are in majority activated[27], (ii) activated NPs promote

quiescence in their immediate neighbors and this is mediated by Notch[27], (iii) N3ICD and *dla* levels are inversely correlated (Fig. 4L), (iv) N3ICD[low] cells are surrounded by higher Notch3 levels than expected (Fig. 3D), and (v) experimentally decreasing *dlaA* in individual cells in situ induces their quiescence[44]. These observations converge towards proposing that Dla is engaged in lateral inhibition with Notch3 to promote NSC quiescence by the induction of high Notch3 signaling levels in contacting cells. *dla*[pos] NSCs do themselves receive a Notch signal, although weaker than their *dla*[neg] counterparts, as indicated by their reactivation frequency upon Notch blockade[44]. This final observation further suggests that Dla is also engaged in cis-inhibition, where its binding to Notch3 in the same cell would decrease Notch3 availability for neighbors.

Although Jag1 is expressed in neurogenic niches in mouse[61,62], the role of Jag1b-Notch3 signaling is addressed here for the first time. Collectively, we show that (i) *jag1b* expression is widespread in NSPCs, (ii) Notch3 signaling and *jag1b* in individual NSPCs positively correlate (Fig. 4G), and (iii) are necessary for each other's expression or activity (Fig. 5C, G). Finally, (iv) lowering Jag1b decreases Sox2 levels. This experiment could theoretically perturb signaling from all Notch receptors, but the prominent expression of *notch3* (Supplementary Fig. S6E) strongly suggest that it prominently affects Jag1b/Notch3 interactions. *sox2* is a direct Notch signaling target in hippocampal NSCs[31]. It promotes the NSPC state in a dose-dependent manner[63,64], an effect already visible when Sox2 dosage is genetically reduced by 3-fold in the adult mouse hippocampus[65]. These results converge to suggest an engagement of Jag1b/Notch3 in an alternative Notch signaling mechanism: lateral induction, the output of which would be to promote stemness. Hey1 may be involved downstream of Jag1 (Supplementary Fig. S7K) (and itself maintains Sox2[26]). The delayed and weak effect of *jag1b* knockdown on *hey1* may suggest additional mechanisms, or that the stability of *hey1* transcripts[66] buffers the effect of a transient *jag1b* knockdown. Formally confirming lateral induction would require assessing N3ICD levels cell-autonomously after down-regulating *jag1b*, for example, after mosaic *jag1b* knock-down.

We previously demonstrated that Notch3 controls quiescence and stemness via distinct downstream effectors[24,26]. We show now that N3ICD levels integrate the combined action of two ligands and signaling modes to control these NSC decision axes. Jagged ligands are generally considered to trigger lower Notch signaling levels than Delta[34,56,67–69], an effect that may result in the activation of different target genes. Here, *jag1b* knockdown reduces Notch3 signaling by 1.6-fold at the population scale (Fig. 5G), close to what is observed upon LY treatment (Fig. 2I), but it is difficult to formally conclude as these interventions are likely partial and, as discussed below, Jag1b/Notch3 signaling also impacts Dla/Notch3 signaling. How individual NSCs decode these two signals and what their respective tipping points are to release NSCs from quiescence or permit lineage progression remain important aspects to be studied. These NSC decisions in vivo are partially linked: for example, self-renewing NSCs activate less frequently than neurogenic NSCs, which are further downstream along lineage progression[44,70]. For self-renewing NSCs to divide yet maintain stemness, it is possible that the quiescence/activation balance is regulated by a different (higher) N3ICD threshold than lineage progression. This would add to the emerging list of examples where graded, i.e., level-based, Notch signaling is used to encode cell (state) diversity[67,71–73]. The tipping points are, however, unlikely to be only quantitative as N3ICD levels in Pcna[pos] cells remain broadly distributed (Fig. 2G) and overlap with their values along lineage progression (Fig. 4L).

## Dla- and Jag1b-mediated Notch3 signaling pattern NSC decisions in physical space

The current work also reveals the distribution of N3ICD levels in physical space. Our statistical method to interpret this distribution is based on a 2D analysis of neighborhoods at the apical plane, as we previously showed that this can capture interactions explaining the spatial NSC activation pattern[27]. This 2D analysis also considers the signaling activity of Dla[pos] delaminating cells as long as they maintain an apical contact, i.e., for the entire duration of delamination, which can take days[52]. It ignores, however, possible interactions occurring deeper in the parenchyma and that would not have visible consequences on the apical surface, as we cannot map individual NSC processes and branches with sufficient precision. The latter remains an important point for future studies.

As discussed above, the spatial surrounding of N3ICD[low], *dla*:GFP[high] cells by the highest N3ICD levels likely reads Dla-mediated lateral inhibition. The broad expression of *jag1b*, and its positive regulatory loop with N3ICD levels suggest, in contrast, that it propagates a minimal level of Notch3 signaling encoding the progenitor cell state across the germinal population. A similar situation was described in the developing inner ear, where Jag-Notch-mediated lateral induction encodes the progenitor state across a prosensory field where Delta-mediated lateral inhibition then selects individual cells for neurogenesis commitment[55–57,74–78]. The superimposition of lateral induction to lateral inhibition was previously modeled, predicting in addition that intermediate levels of Jagged-mediated lateral induction reinforce lateral inhibition[58,59,79]. Our spatial analyses of N3ICD signaling upon Jag1b invalidation interestingly fit these predictions: the difference between the highest and lowest N3ICD levels is decreased (Fig. 5G) while a pattern structured by lateral inhibition is retained (Figs. 5J, 6). Several interpretations for this phenomenon have been proposed, such as cis-inhibition of Notch by Jagged (which, when released, would free Notch to interact with Delta presented in trans)[58], or blockade of cis-inhibition of Notch by Delta via the formation of Jagged-Delta dimers[80,81].

Together, we propose that the different expression and signaling modes of the two ligands superimpose and refine local and global information, patterning Notch3 signaling at cell- and tissue-scale. Because N3ICD levels read the position of individual NSC along their two main axes of temporal progression, these cell-cell interactions may coordinate the individual trajectories of otherwise asynchronous NSPCs. In particular, Dla[pos] signaling cells are generated by lineage progression and are transient[44]; thus, the Dla[pos] pattern is spatially dynamic and includes an intrinsic temporal value. We speculate that the combined spatial vs spatiotemporal activities of Jag1b and Dla, respectively, encode the dynamic homeostasis of the pallial NSPC population, maintaining individual cell trajectories in spatiotemporal equilibrium.

## Methods
### Zebrafish
All the reagents, materials and resources used in this study are listed in Supplementary Table 1. All the zebrafish (*Danio rerio*) used in this study have AB or Casper[82] (*roy*[−/−];*nacre*[−/−]) genetic background. They are kept at 28.5 °C in a pH 7.4 water, on a cycle of 14 hours light and 10 hours dark, in 3.5 L tanks at a maximal density of 5 fish per liter. Feeding is performed three times a day with rotifers until 14 days post-fertilization and with standard commercial dry food (GEMMA Micro from Skretting) afterwards. Embryos and larvae were staged and raised according to ref. 83. The developmental stage is indicated for each experiment. Adult fish were between 3 and 4 months old. Fish from the same parents were used in the same experiment. Both males and females were included in the analysis since no difference in brain development and NSCs homeostasis was observed in the past. Fish were euthanized in ice-cold water

(temperature below 4 °C for 10 min, according to a special dispensation and following the guidelines of the Ministry of superior education, research and innovation.

The two transgenic lines generated for this study are detailed in the next section. Tg(dla:GFP)[45] and notch3[fh332/+] line[24] were used for crossings.

## Zebrafish TgBAC(notch3:notch3-GFP-P2A-nlsRFP)[ip13Tg] line generation

Genomic zebrafish BAC CH211-69E21 was modified to insert a "Tol2L-AmpR-Tol2R-polyA-βcrystallin:eCFP" cassette at the loxP511 locus to enable transposase-mediated recombination and selection via the βcrystallin promoter. The BAC was further modified by homologous recombination to insert "linker-GFP-P2A-nlsRFP" sequence in position + 6823 of notch3 coding sequence, in frame with the exon33. The construct "linker-GFP-P2A-nlsRFP" was obtained using the NEBuilder HiFi DNA Assembly Master Mix (NEB, Cat#E5520S) to assemble a short linker (translating into "GVG" peptide), a GFP sequence (cloned by PCR from the plasmid p3E-IRES-EGFPpA – Tol2kit[84] #389), a ribosome-skipping P2A sequence[85] and a nlsRFP sequence (gift from Holley SA). This construct was then recloned using KpnI and SacI restriction sites into a modified pBluescript vector, along with two sequences in 5' and 3', homologous to the insertion site on the notch3 sequence, to increase recombination efficiency. A Zeocin cassette oriented 3'-5' was inserted downstream the stop codon of nlsRFP, to facilitate selection of recombined clones. Recombined clones were selected, verified by PCR and sequencing of the insertion sites, and by EcoRI restriction profile (Agate bioservices, Bagard, France - see Supplementary Fig. S1C for scheme).

The TgBAC(notch3:notch3-GFP-P2A-nlsRFP)[ip13Tg] line was obtained after microinjection of approximately 1 nl in one-cell stage zebrafish embryos of a solution containing the modified BAC (80 ng/ml) together with transposase-capped RNA (40 ng/ml). Founder adults were screened by backcrossing in wild-type lines and monitoring of CFP expression in the eyes of F1 progeny. The TgBAC(notch3:notch3-GFP-P2A-nlsRFP)[ip13Tg] line was maintained in the Casper double mutant (roy[-/-];nacre[-/-]) background. Fish carrying the transgene were routinely selected at 48hpf based on fluorescence expression.

## Zebrafish TgKI(notch3:notch3-mAG-P2A-nlsRFP)[ip14Tg] generation

**Cas9 target sequences** were identified using the CRISPOR tool[86,87] for genome editing (see Supplementary Fig. S1D). Custom crRNAs corresponding to each sequence and universal tracrRNA were ordered from IDT (Alt-R CRISPR-Cas9 system), annealed by heating equimolar amounts to 95 °C for 5 min and then cooling to room temperature. Final guideRNA (gRNA) duplex concentration was 60 μM. Lyophilized Cas9 nuclease (Labomics, Cat#Cas9-TOO-250) was reconstituted to 30 μM in high-salt buffer[88] (750 mM KCl, 50% glycerol).

**Ribonucleoprotein complex (RNP)** was assembled on injection day by mixing Cas9 (2 μL, 30 μM), gRNA duplex (1 μL, 60 μM), and nuclease-free water (2 μL). Complexes were incubated at 37 °C for 5 min, then kept on ice. To test gRNAs efficiency, 1 nl of the RNP complex was injected into fertilized zebrafish eggs at the one-cell stage. At 48hpf, genomic DNA was extracted from 20 embryos per condition using Phire Animal Tissue Direct PCR Kit (ThermoFisher Scientific, Cat#F140WH). Target region on notch3 gene was PCR-amplified (forward primer: 5'-GGTGCACAGCAGTATCCTG-3' and reverse primer: 5'-GCGGATACCGGCAGATGG-3') and cutting efficiency assessed using T7 endonuclease I assay according to manufacturer's instructions (NEB, Cat#M0302S) (see Supplementary Fig. S1E). gRNA4 sequence (5'-GGGGTAATCCTCTGGGCCTGCGG-3') was selected to generate the knock-in line.

**A donor vector** was designed to insert a 1530 bp "linker-AzamiGreen-P2A-nlsRFP" cassette in frame with the exon 33 of the notch3 gene. P2A and nlsRFP sequences were obtained as above; CoralHue monomeric AzamiGreen sequence was cloned by PCR from pmAG1-MN1 plasmid (MBL, Cat#AM-V0033). The pCS2P backbone was modified by PCR to remove the CMV promoter. Long homology arms (around 900 bp for the left arm and 800 bp for the right arm) were designed starting from the Cas9 genomic cutting site specific for gRNA4, to enhance recombination frequency. Homology arms were synthesized as gBlocks (IDT) and contained silent mutations in the Cas9 recognition sequence to prevent re-cutting after integration. Cas9 target sites specific for gRNA4 were positioned flanking both homology arms in opposite orientations to allow simultaneous linearization of donor and genomic DNA with the same guideRNA. A 17 amino acid linker sequence was included between Notch3 and AzamiGreen to prevent folding interference[89]. The four fragments (modified vector backbone, left homology arm, "linker-AzamiGreen-P2A-nlsRFP" insert, right homology arm) were assembled at equimolar ratios (0.05 pmol each) using NEBuilder HiFi DNA Assembly Master Mix (NEB, Cat#E5520S).

**On injection day**, the ribonucleoprotein complex was assembled and injected as above, with the donor vector (2 μL, 500ng/μL) replacing water. At 48hpf, individual embryos were screened for successful integration efficiency (see Supplementary Fig. S1F). Genomic DNA extraction was performed using Phire Animal Tissue Direct PCR Kit (ThermoFisher Scientific, Cat#F140WH). Target regions spanning the integration sites were PCR-amplified (for 5' integration: forward primer: 5'-CATGGACAGACTCCCTCGAG-3' and reverse primer: 5'-GGTGCGCCTTCAGTGACG-3'; for 3' integration: forward primer: 5'-GCCTACAAGACCGACATCAAG-3' and reverse primer: 5'-CAGCTCTGGCCTTTGTAAGTC-3'), purified, A-tailed, and TA-cloned (StrataClone PCR cloning kit, Agilent Technologies, Cat# 240205) for sequence verification to confirm precise integration and absence of indels. Fish batches showing correct integrations were screened at 3mpf for germline transmission: this involved fluorescence observation of offspring between 48hpf and 72hpf, followed by sequencing validation of integration sites at 3mpf using tail-extracted DNA and the methodology described above. Once established, the TgKI(notch3:notch3-mAG-P2A-nlsRFP)[ip14Tg] was maintained in Casper double mutant (roy[-/-];nacre[-/-]) background. Fish carrying the transgene were routinely selected at 48hpf based on fluorescence expression.

## Genotyping

Adult fish genotyping was performed using tail samples. Before tissue collection, fish were anesthetized by immersion for 1 to 2 minutes in water containing 0.01% MS222 (Sigma-Aldrich, Cat# A5040). Tail clips were collected, and DNA was extracted using the Phire Animal Tissue Direct PCR Kit (ThermoFisher Scientific, Cat#F140WH) according to the manufacturer's instructions.

**TgKI[n3-AG/+] versus TgKI[n3-AG/n3-AG]** genotyping was performed using primers positioned outside the insertion site (forward primer: 5'-CATGGACAGACTCCCTCGAG-3' and reverse primer: 5'-GCGGATACCGGCAGATGG-3'). GoTaq- mediated PCR (GoTaq G2 Hot Start Master Mix – Promega, Cat#M7422) was conducted on a pre-heated thermocycler according to the following protocol: 95 °C x 2', [95 °C x 30' – 54 °C x 30' – 72 °C x 3'] repeated for 35 cycles, 72 °C x 5'. PCR products were analyzed by electrophoresis on 1% agarose gel to discriminate the alleles according to their size: 1.5 Kb for the WT allele and 3 Kb for the knocked-in allele.

**TgBAC[n3-GFP/+] crossed with notch3[fh332/+]** offspring genotyping was performed as previously published[24], on fish selected for BAC expression at the embryonic stage. After PCR product digestion and electrophoresis on 3% agarose gel, different genotypes were discriminated according to band intensity: the presence of only the upper band indicated 3 functional notch3 copies, two bands of the same intensity indicate 2 copies of notch3[fh332] and 1 notch3 WT copy. An

upper band of high intensity and a fainter lower band indicated 1 copy of *notch3*[fh332] and 2 WT *notch3* copies.

## in situ hybridization (ISH)

DIG-labeled probes for *notch3*[23], *gfp* (obtained from an in-house plasmid) and *AzamiGreen* (PCR cloned from the KI donor vector described above [forward primer: 5'-ATGGGTGAGTGTGATTAAACCAGAG-3' and reverse primer: 5'-CTTGGCCTGACTCGGCAGC-3'] and inserted into the pSCA vector using the StrataClone PCR cloning kit) were produced incubating 5 μL of linearized DNA (1 μg), 2 μL DIG-NTP labeling mix (Merck, Cat#11277073910), 1 μL RNase inhibitor (Takara Bio, Cat#2313 A), 4 μL 5X transcription buffer, 2 μL RNA polymerase (T3 or T7, Promega, Cat#P2083/2075), and 6 μL nuclease-free water at 37 °C for 3 h. Following transcription, 1 μL DNase I (Merck, Cat#4716728001) was added and incubated for 15 min at 37 °C to remove template DNA. The probe was purified using G-50 Micro Columns (Merck, Cat#GE28-9034-08).

28hpf embryos from WT, TgKI[n3-AG/+] or TgBAC[n3-GFP/+] lines (sorted based on fluorescence) were dechorionated using pronase and fixed overnight in 4% PFA (Paraformaldehyde, ThermoFisher Scientific, Cat#15710) at 4 °C, then dehydrated through a graded methanol series (25%, 50%, 75%, 100% in PBT) and stored at −20 °C. On Day 1, samples were treated with 3% H₂O₂ diluted in methanol for 20 minutes at Room Temperature (RT), rehydrated through the reverse methanol/PBT (PBS [Phosphate-Buffered saline, ThermoFisher, Cat#10649743] with 0.1% TWEEN-20, Merck, Cat#P9416) series and treated with proteinase K (ThermoFihser Scientific, Cat#EO0491) at 10 μg/ml for 4 min, then post-fixed in PFA 4% for 20 min at RT. For chromogenic ISH, hybridization and revelation of the whole embryos were performed as previously published[23]. Another post-fixation step in PFA 4% for 20 min to 1 h was performed before imaging with an Axio Zoom V16 stereo microscope (Zeiss). For fluorescent ISH, probe hybridization was performed as described above, followed by overnight incubation at 4 °C with an anti-Digoxygenin antibody conjugated with Peroxidase. Signal development was achieved using the RNAscope kit and OPAL-570 fluorophore at 40 °C for 30 min. Images were acquired using a BC43 spinning disk confocal microscope (Andor).

## Immunohistochemistry

Fish between 3 and 4 months old were euthanized as described above, and whole brains were dissected in cold PBS, fixed in 4% PFA at RT for 2 h and dehydrated through sequential washes in 25, 50, 75 and 100% methanol (diluted in PBT). Brains were stored at −20 °C for at least one night. After rehydration in reverse methanol series, antigen retrieval was performed using HistoVT One solution (Nacalai Tesque, Cat#06380-05) at 65 °C for 1 h (for anti-BrdU antibody in Supplementary Fig. S3A, this step was replaced by a HCl 2 M treatment for 20 minutes at RT). Samples were soaked in blocking solution (5% Goat Serum Donor Herd [Merck, Cat#G6767], 0.1 % Triton-X100 [Merck, Cat#X100], 0,1 % DMSO [Merck, Cat#D8418] in PBS) for at least 1 h at RT. Primary and secondary antibodies were diluted in the blocking solution and incubated overnight at 4 °C (except for Alexa Fluor-directly conjugated anti-ZO1antibodies, that were usually incubated between 1 and 2 h at RT). Samples were washed several times in PBDT at RT after each antibody incubation. When needed, goat serum was replaced with 4% of natural mouse serum (Abcam, Cat#ab7486) in the blocking solution. Dissected telencephala were mounted in PBS on slides using 0.5 mm-thick holders (iSpacer, Nikon, Cat#2SUN1016). The slides were sealed using Valap, a mixture of petrolatum jelly (Merck, Cat#16415), paraffin wax (Merck, Cat#76242), and lanolin (Merck, Cat#L7387).

For embryos or larvae, brains were previously cut either in 10 mm-thick sections with a Leica CM3000 cryostat (for 29-30hpf embryos, beforehand included in 7.5% gelatin–15% sucrose solution and frozen by immersion for 30 s in a bucket containing isopentane and maintained in liquid nitrogen to reach a temperature between −60 °C and −80 °C) or in 50 mm-thick sections with a Leica VT1000 S vibratome (for 8dpf larvae). Sections were then processed as above and mounted on slides sealed with Aqua-Poly/Mount (Polysciences, Cat#18606).

## RNAscope

The protocol for RNAscope fluorescent in situ hybridization (FISH) (Advanced Cell Diagnostics, which is part of BioTechne, Cat#323100) was adapted to whole-mount fish brains[50], and all steps were performed in 1.5 ml tubes. Adult brains were dissected, fixed and dehydrates as described in the immunohistochemistry protocol. On Day 1, after rehydration, a pigment removal step was performed by incubating tissues in a water solution containing 3% H₂O₂ (Merck, Cat#216763), 5% formamide (Merck, Cat#47671) and 0.5X SSC (ThermoFisher Scientific, Cat#10515203) and exposing them to direct light for 10 minutes. After washing, tissues were incubated in RNAscope's proprietary diluent buffer from Bio-Techne (Cat#324301) at 40 °C for at least 2 hours. Primary probes diluted 1:50 in diluent buffer and prewarmed for 10 min at 40 °C were then incubated with the brains overnight at 40 °C with gentle shaking. On Day 2, after a series of washes and amplification steps with Amp1, Amp2 and Amp3 buffers, samples were treated with the appropriate HCR reagent (-C1, -C2 or -C3 in accordance with the probe used) at 40 °C for 15 min, followed by incubation with OPAL dyes (480, 570 or 650) at 40 °C for 30'. The reaction was blocked with HRP-blocker at 40 °C for 15 min. Sequential detection with new HRP-dye combinations was performed if required, allowing visualization of up to three different colors. To assess specificity and exclude background noise, negative control probes in each channel were tested prior to experiments. When necessary, immunohistochemistry was then performed on the same samples, following the protocol described above except for the antigen retrieval step in HistoVT One that was omitted.

## Image acquisition

Image acquisition of brains was obtained on an LSM980, LSM700 or LSM710 confocal microscope (Zeiss) using a 40X oil objective (Plan-Apochromat 40 × /1.3 Oil M27 − Effective NA between 1.3 and 1.4) and processed with the ZEN software (version Blue or Black, Zeiss). For adult brains, the dorso-medial part (Dm)[90] was imaged for each telencephalon. Laser power (from 1% to 6%), detector gain (from 600 to 900) and Z-steps were optimized for each experiment and maintained constant for all the samples of the same experiment. Bit depth for all the images acquired was 16 bit, and size 1024 × 1024 pixels (except for Supplementary Fig. S2D [where Rab5 was 1894 × 1894 pixels] and S2E [where Rab7 was 1893 × 1918 pixels and Lamp1 was 1894 × 1898 pixels]). Scan zoom was set between 0.8X and 1.2X, except for Supplementary Fig. S2E where the scan zoom was 2.5X. 3D renderings were generated using Imaris software (versions 8 and 9, Bitplane). Vertical plane images were extracted when needed.

Supplementary Fig. S2C was recorded on a living embryo (dechorionated, anesthetized as above and maintained in 1.2% low-melting agarose) with the LSM700 confocal microscope, focusing on one Z plane and leaving the objective open for around 1 minute.

Supplementary Fig. S2B was obtained on a living embryo (dechorionated, anesthetized as above and maintained in 1.2% low melting agarose) with a Z1 Lightsheet microscope (Zeiss), a 40x W Plan-Apochromat objective with water immersion and a numerical aperture of 1.0.

## BrdU pulse

Dechorionated embryos were put in a solution containing 10 mM BrdU (Merck, Cat#B5002), 15% DMSO and embryo medium, left on ice for 20 minutes, then transferred at 28 °C for 20 min to recover. Embryos were euthanized, fixed and immunostained as described above.

## LY-411575 treatment

Inhibition of Notch signaling was performed using the LY-411575 γ-secretase inhibitor (Merck, Cat#SML0506), prepared and stored as a 10 mM stock solution in DMSO and applied in the fish swimming water at a final concentration of 10 µM for 24 h. Control fish were treated with the same final concentration of DMSO solvent.

## Morpholino (MO) injection

To selectively knock down *jag1b* in the whole NSPC population, an in vivo-MO construct provided by the Gene Tools company was injected into the adult pallium in vivo. The ATG-directed jag1bMO sequence (5′-CTGAACTCCGTCGCAGAATCATGCC-3′) was previously validated[91], and a generic controlMO sequence (5′-CCTCTTACCTCAGTTA-CAATTTATA-3′) was used to compare the effects. vivo-MO were injected at a final concentration of 0.125 mM directly in the cerebrospinal fluid of fish anesthetized by immersion for 1 to 2 min in water containing 0.01% MS222. Brains were dissected and fixed as above, 1 to 3 days after injection.

## Identification of RBPj sites on intron 1 and 2 of *jag1b* gene

The intron 1-2 and 2-3 sequences of the *jag1b* gene (ZDB-GENE-011128-4) were used as input to scan for transcription factor binding site motifs in the Animal Transcription Factor Database v4.0 (https://guolab.wchscu.cn/AnimalTFDB4/#/).

## Quantification and statistical analysis

**Nuclear signal quantification**. A semi-automatic image analysis pipeline was developed for this study using Imaris (Bitplane) software to quantify Notch3 activity in zebrafish transgenic lines. Individual imaging fields (100–200 cells each) were independently segmented and treated as discrete datasets to ensure homogeneity. Edge effects were minimized by excluding cells positioned at the image periphery. The pipeline employed two distinct approaches for nuclear segmentation depending on the marker used.

When using the nlsRFP signal for nuclear identification, a pre-processing step was required to avoid strong red background noise that could impact segmentation. This involved: using the Surfaces module on the red channel, including all the strong and not specific red dots (settings: smooth enabled with surface grain size = 0.2 mm; method = background subtraction with manual threshold; diameter of the largest sphere = 0.5 mm), placing signal intensity within these surfaces to zero, and generating a new cleaned channel where these surfaces are masked. The Surfaces module was then applied again on the cleaned red channel (settings: smooth enabled with surface grain size = 0.45 mm; method = background subtraction with manual threshold; diameter of the largest sphere = 1.29 mm; seed points diameter = 3 mm, intensity based) to identify and segment 3D nuclear objects. If needed, manual correction of individual nuclei was conducted to avoid errors.

When using the Sox2 signal for nuclear segmentation, the first cleaning step was unnecessary due to the high specificity and clarity of the anti-Sox2 antibody signal (settings: smooth enabled with surface grain size = 0.45 mm; method = background subtraction with manual threshold; diameter of the largest sphere = 1.29 mm; seed points diameter = 3 mm, intensity based).

After segmentation, the following quantitative parameters were extracted from each nucleus across all channels: spatial coordinates (position of the center of each surface for spatial statistical analysis), nuclear volume (3D volume measurement of segmented nuclei), sum (total pixel intensities within each nuclear surface) and concentration (conc°) (average pixel intensity normalized by nuclear volume). For Notch3 activity assessment, sum intensity represents the absolute amount of nuclear Notch3 receptor, while conc° represents the concentration of Notch3 within the nucleus. Since the most biologically meaningful parameter for Notch3 activity quantification remains undetermined, both measurements were systematically analyzed to provide a comprehensive assessment of receptor nuclear levels.

## mRNA quantification

mRNA dots revealed after RNAscope were quantified using Imaris (Bitplane) software. The Spots module was used to identify dots in the chosen channel (settings: 0.5 mm estimated diameter, background subtraction, quality above 1000), which were then manually assigned to individual cells, except for Fig. 5C, D and Supplementary Fig. S7I, where the total number of dots on the image was divided by the number of cells with an apical area.

## Apical area and cytoplasmic signal quantification

Apical area was automatically calculated using Napari software (running on Python 3.9) and the Fishfeats plugin (in-house code based on Epyseg[92] and big-FISH[93], which will be published elsewhere). Zo1 signal was selected to identify apical membranes. Segmentation of individual cell areas was corrected by hand if needed. After apical area segmentation, each cell was manually assigned to its identity as determined on Imaris software, to enable correlations between Napari and Imaris quantification datasets.

Cytoplasmic GFP signal in Fig. 4L was also quantified using Napari software, after apical area segmentation: the mean value of GFP on a Z thickness of 1 under the apical surface was extracted for each cell, and separated in three categories (low, medium and high).

## Statistical analysis of quantifications

All statistical analysis were performed using R or the Excel add-in Analysis ToolPak. All the tests used are included in the figure legends, along with *p*-values and n (number of cells per condition, collected from 1 to 5 fish, or number of hemispheres analyzed, as indicated). Experimental data are presented using different visualization methods depending on the purpose: beeswarm plots and scatter diagrams (individual dots representing single cells and one black dot representing the mean value), histograms, bar plots (representing mean ± sd) or boxplot (showing median, mean, interquartile range, whiskers extending to the minimum and maximum values, and outliers). Z-scores (Standard scores = [(value − mean of the population)/standard deviation]) were calculated when needed, to be able to normalize and achieve uniform inter-samples comparisons. For correlations of Z-score values and other parameters such as apical area or mRNA dots (as in Fig. 4G, I, M, N), range similarity of apical areas or dot counts across samples was confirmed prior to analysis. The Shapiro-Wilk test was used to verify normality. When data were not normally distributed, the variance was not homogeneous and/or the sample size was too small, non-parametric tests were used: two-sided Wilcoxon rank sum test to compare two samples (as in Figs. 1J, 2F, G, I, 4D, E, K, Supplementary Fig. S7F and Fig. 5) and Spearman's rank correlation to measure statistical dependence between two variables (as in Fig. 4C, G, I, M, N and Supplementary Figs. S6A, B, S7D, E). When parametric tests were appropriate, a two-sided Student's *t* test was used to compare 2 samples (Supplementary Fig. S3G, H) and ANOVA or Welch's ANOVA tests were employed to compare three or more groups with (Fig. 1K) or without equal variances (Figs. 2K, 4L and Supplementary Fig. S3D, E). Chi-Square test of independence ($X^2$ test) was used to compare the frequency distribution across different groups (Fig. 2D, E). Statistical significance was defined as $p < 0.05$. Data presented with asterisks indicate significance levels: *$p < 0.05$, **$p < 0.01$, and ***$p < 0.001$.

## Spatial statistics

**Summary statistics: marked *L*-function**. The main goal is to study the correlation between the spatial structure of the adult pallial ventricular cells and their levels of N3ICD. We make use of second-order summary statistics, which are used for describing correlation patterns between points in spatial settings.

For this, we model this tissue as a *marked* point process: we represent the spatial positions of the nuclei of the cells by the coordinates $(x_i)$, and to each point in space, there is an associated *mark* noted $(m_i)_i$ (for instance, N3ICD sum, N3ICD mean, or Apical Area). The mark represents any numerical value that is related to a physical position.

To describe the correlation between marks by taking into consideration the relative position of points, it is common practice to use the *marked K-function* denoted $K_m$ [94]. Adapted from Ripley's *K*-function[95], for any given radius $r$, this function counts the pair of points that are within a distance of $r$ weighted by a function $f$ of their corresponding marks. In order to compare the *K*-function between different point processes, this quantity is normalized by both the average number of points per unit squared and by the expected value of the marks.

In our study, we focus on the marked version of Bessag's *L*-function, noted $L_m = \sqrt{(K_m)/\pi}$ (Figs. 3D, 6C and Supplementary Fig. S4D). This corresponds to a variance-stabilized version of $K_m$ which tends to display better statistical properties.

To avoid edge issues when computing the empirical *L*-function $\hat{L}_m$ from our data, we add an edge correction term to this quantity through translation correction as described in Section 7.4.5 in Baddeley et al.[96]. Technical details on theoretical guarantees, statistical inference and numerical computation of the $K_m$ and the $L_m$ functions can be found in ref. [97] and ref. [96].

**Benchmarking spatial randomness.** The function $\hat{L}_m$ alone can help understanding specific spatial patterns linked to the studied marks, such as clustering or repulsiveness, but its exact interpretation can be difficult without comparing it to a reference model. Therefore, we decide to compare the observed function $\hat{L}_m$ to a benchmark $L_0$ representing a model without correlation between marks and spatial locations. Then, if for a certain radius $r$, we observe that $\hat{L}_m < L_0$ (respectively $\hat{L}_m > L_0$), then this suggests that the marks of all points in a radius of $r$ from a typical point in space are "lower" (resp. "higher") than expected if there was no correlation between space and marks.

The first approach would be to define a mathematical model able to simulate the spatial location of cells in the adult pallium along with theoretical N3ICD values that would be biologically close (in a sense) to what would be expected of a tissue where no correlation exists between these two quantities. Such a model would need to take into account the particular structure of an epithelial tissue (minimum distances between cells, fully populating the space in terms of apical area) and the mechanism generating the different N3ICD values of each cell. This is the first hurdle. Second, it would be necessary to either have a closed-form expression of its theoretical function $L_0$ or to be able to simulate such a process to obtain an empirical version of the function. In practice, this is a very difficult challenge.

To define this benchmark model of spatial uncorrelation, we turn to permutation methods for approximation. The goal is to generate an empirical version $\hat{L}_0$ of what would be this $L_0$ function using only the observed data. For this, we simulate 100 "random" scenarios where the locations and marks are uncorrelated. For each simulation, we compute its empirical marked $L$ function, and by averaging them we obtain $\hat{L}_0$.

It remains to define how to simulate such scenarios. Remember that we observe a set of locations $(x_i)$ and associated marks $(m_i)$. We simulate "random" scenarios by permuting the marks associated to each location as follows:

1. We fix the spatial locations, so the nuclei of the cells are assumed to be always in the same place.
2. We randomly exchange the order of the marks $(m_i)_i$ with a permutation $\sigma$ to obtain $(m_{\sigma(i)})_i$.

3. We then associate to each point $x_i$ its "new" mark $m_{\sigma(i)}$.

Note that if the observed marks truly come from this "random" scenario, then the observed pattern is exchangeable with the simulated pattern.

Averaging 100 different permutations allows to obtain the "expected" value of the marked $L$ function, that can be then used for comparison.

**Hypothesis testing and simulation envelopes.** To account for the inherent randomness of any probabilistic model, we carry out a hypothesis testing procedure to compare $\hat{L}_m$ and $\hat{L}_0$ and to make any interpretations. The null hypothesis we test is:

$$\mathcal{H}_0 : L_m = L_0,$$

If $\mathcal{H}_0$ is true, this suggests that there is no correlation between the spatial structure of nuclei and their corresponding N3ICD values. The chosen procedure is a deviation test, as it was first proposed in ref. [98]. A well-detailed explanation is also present in Myllymäki et al.[99] that we summarize here:

1. Let $R > 0$, we compute the maximal deviation of our observation with respect to the theoretical value $\hat{L}_0$:

$$u = \max_{r \in [0,R]} |\hat{L}_m(r) - \hat{L}_0(r)|. \tag{1}$$

2. For a set of $s$ simulated realizations of the benchmark model, we estimate their respective marked L functions $\hat{L}_i$.
3. We compute the deviations:

$$u_i = \max_r |\hat{L}_i(r) - \hat{L}_0(r)|. \tag{2}$$

4. The estimated *p*-value of our test is:

$$\hat{p} = 1 - \frac{1}{s+1} \sum_{i=1}^{s} \mathbf{1}(u_i < u). \tag{3}$$

In practice, we obtained the simulations in step 2 by considering the same simulation procedure as before.

A limitation with this deviation test is that while it allows to test hypothesis $\mathcal{H}_0$, it does not inform about the specific radius at which rejection occurs, nor if the rejection is due to $L_m > L_0$ or $L_m < L_0$ or a mix of both effects. For this, we add a visualization of the *global envelope tests* as originally proposed in Ripley[95], which correspond to Figs. 3D, 6C and Supplementary Fig. S4D. We represent the "deviation" $\hat{L}_m - \hat{L}_0$ of the observed value from the expected "uncorrelated" function (as a continuous line). Then we compute the maximal simulated deviation:

$$D_{\max} = \max_i u_i \tag{4}$$

and we represent the rejection thresholds $y = D_{\max}$ and $y = -D_{\max}$. If the "deviation" curve goes above $D_{\max}$ (resp. below $-D_{\max}$) at a radius $r$, it suggests that the marks of all points within a radius $r$ of a typical point is higher (resp. lower) than expected if marks and space structure were uncorrelated. This way, any rejection of hypothesis $\mathcal{H}_0$ can be interpreted as the presence of correlation between cells' N3ICD values at a distance smaller or equal to $r$.

In our simulations, we choose $s = 99$ in order to have a confidence level of 99% on any hypothesis rejection.

**Practical considerations.** In this section, we specify the different choices made for our data.

- For the 3 studies carried out, we have different numbers of samples available (2 hemispheres for TgKI$^{n3AG/+}$, 1 hemisphere for TgBAC$^{n3-GFP/+}$, 6 hemispheres for each *ctrlMO* and *jag1bMO*). As the marked *L*-functions are normalized by the local density of points and mark values, all functions can be directly compared. Therefore, to obtain the final function $\hat{L}_m$ of an experiment, we pool the estimations of each sample before implementing the hypothesis testing and envelope curves.

- As our study focuses on describing the correlations surrounding N3ICD$^{low}$ cells, and as these values may be highly variable in distribution between individual fish, the groups are defined locally. The N3ICD$^{low}$ cells correspond to the 20% of cells with lower N3ICD activity for the given sample.

  Once the group N3ICD$^{low}$ is defined, all the marked *L*-functions described before are computed by considering as "center" points these cells. In practice, for each position $x_i$ of an N3ICD$^{low}$ cell, we compute the mark-weighted sum of **all** surrounding cell at a given distance $r$.

  When permuting marks to approximate the benchmark model, the marks of all points (including from N3ICD$^{low}$ cells) are randomly exchanged.

- As mentioned in the description of the marked *K*-function, it is important to choose a function $f$ that defines the *weight* of the marks. More precisely, if two points $x_i$ and $x_j$ are at a distance smaller than a studied radius $r$, their contribution to the sum is equal to $f(m_i, m_j)$. Different choices of $f$ are used in the literature, many of which are described in Section 5.3.3 in Illian et al.[97]. Although we consider different options in our study, the presented results correspond to the function $f(m_i, m_j) = m_j$. This function allows to represent the total mark value surrounding a cell at a position $x_i$ for a radius $r$.

  This choice is justified because we decided to focus on studying N3ICD$^{low}$ cells, and so we want to focus on their surrounding signal.

## Ethics

The animal study protocol was approved by the Ethics Committee n°39 of Institut Pasteur (authorization #36936, April 26th, 2022) and DDPP-2021-021 of the Direction Départementale de la Protection des Populations de Paris.

## Reporting summary

Further information on research design is available in the Nature Portfolio Reporting Summary linked to this article.

## Data availability

Lead contact: laure.bally-cuif@pasteur.fr. Materials availability: all materials and transgenic lines generated in this study are available upon request to laure.bally-cuif@pasteur.fr. Source data are provided in this paper.

## Code availability

The spatial statistics analysis was coded in Python. Computing the $K_m$ and $L_m$ functions along with the hypothesis testing procedures were entirely implemented by leveraging classic functions from pandas, NumPy and Scipy libraries and parallelization was performed using the multiprocessing package. All data (raw and processed), codes to compute the quantities and to generate all images in Figs. 3, 6 and Supplementary Fig. S4, and a Readme can be found in https://zenodo.org/records/18244815 (ref. 100).

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

## Acknowledgements

We are greatly indebted to François Schweisguth for discussions and advice along this work, to Felix Cheysson for advice on spatial statistics, to Neetu Gupta-Rossi for the gift of antibodies against endocytic pathway components, to Sébastien Bedu and Nathan Guibert for expert fish care, to members of the LBC team for support and discussions, and to M. Wappner for his critical reading of the manuscript. This research was funded by the ANR (Labex Revive ANR-10-LABX-0073 to L.B.-C.), La Ligue Nationale Contre le Cancer (LNCC EL2019 BALLY-CUIF to L.B.-C.), the Fondation pour la Recherche Médicale (EQU202203014636 to L.B.-C.), the European Research Council (ERC SyG PEPS 101071786 to L.B.-C.), CNRS, and Institut Pasteur. L. Degroux was a recipient of an Institut Pasteur work-study contract.

## Author contributions

Conceptualization: L.B-C., N.D., and S.O.; Methodology: L.D., B.R., M.MH., and S.O.; Writing original draft: L.B-C.; Review and editing of original draft: all authors; Supervision: L.B-C. and N.D.; Project administration: L.B-C.; Funding acquisition: L.B-C. All authors have read and agreed to the published version of the manuscript.

## Competing interests

The authors declare no competing interests.
