## [Transparent Peer Review file · Nature Communications]

Jagged-mediated lateral induction patterns Notch3 signaling within adult neural stem cell populations

Corresponding Author: Dr Laure Bally-Cuif

Version 0:

Reviewer comments:

Reviewer #1

(Remarks to the Author)

This manuscript uses expression of functional Notch3 fluorescent variants in BAC or knockin contexts to analyze the circuitry of Notch3 signaling in the zebrafish brain as it relates to neural stem cell quiescence. The authors leverage quantitative imaging, manipulation of Notch signals with inhibitors or Jag1b-directed morpholinos, and careful data analysis to draw the conclusion that Notch3 in NSCs is responding to Delta in neighboring cells, which results in the induction of Jag1b in the Notch3-high cells to maintain and reinforce the NSC fate in signal-receiving cells. The authors clearly indicate when their analyses provide robust results and when they show trends or outcomes that are not statistically significant. Overall, the work is thorough and support the authors' conclusions.

Additional comments

1. The authors note that previous modeling predicts that superposition of lateral induction on lateral inhibition results in more robust pattern formation. It would be interesting to include a mathematical model showing how the robustness of pattern formation varies as a function of the lateral induction signal strength.
2. The knockin of the fluorescent label onto Notch3 allows the authors to distinguish Notch3 signaling from signals due to activation of other Notch proteins in the same cell types. However, use of gamma secretase inhibitors results in the silencing of all Notch signals in treated cells. Can the authors comment on how the inhibition of all Notches might affect the interpretation of their results in Figs. 5A-5D?
3. The authors acknowledge the possibility that the half-life of the fluorescent fusions may differ from the wild-type Notch3 allele because the fusion sites alter or remove the PEST sequence implicated in protein degradation. How might longer persistence of the mutant N3ICD affect interpretation of the data? Is it possible to examine the difference in half-life of the wild-type and knockin N3ICD experimentally by monitoring the rate of decay of the two alleles after gamma secretase treatment?

(Remarks on code availability)

Reviewer #2

(Remarks to the Author)

This manuscript by Sara Ortica and colleagues presents a detailed and quantitative study on how Notch3, in response to its DeltaA and Jagged1b ligands, controls lineage progression during neurogenesis in the zebrafish pallial brain. The authors have developed and validated a set of genetic tools that enable precise monitoring of Notch3 activity, which they used to dissect the spatial control of this signaling pathway. The innovative tools and rigorous statistical analysis together yield novel insights into how Notch signaling is patterned in the zebrafish brain. The manuscript also provides a careful characterization of *dla* and *jag1b* expression, followed by functional experiments that test their respective contributions to Notch3 regulation. These findings led the authors to propose a model in which lateral inhibition and lateral induction are

integrated to fine-tune Notch3 activity, thereby influencing the balance between neural stem cell quiescence and activation. This model provides a new perspective on how Notch3 regulates neurogenesis in the zebrafish pallial brain and offers a framework that is likely to be relevant across vertebrate systems.

Overall, the manuscript is a strong contribution that combines technical innovation with biological insight. The work advances our understanding of neurogenesis in the vertebrate pallium and will be of broad interest to researchers studying neural stem cell regulation and Notch signaling.

The paper is well organized, although at times the text is difficult to follow due to insufficient explanation of certain concepts. Addressing the following points might substantially improve the manuscript's clarity and impact.

Major Points

1. Apical localization and apical size: In several analyses, the apical localization of cells or the size of their apical domain is treated as an important parameter. However, the functional significance of this parameter is never explained. A brief description of why apical localization/size is relevant for the analyses presented here would greatly help the reader. Although this was addressed in a previous publication from the lab, a short explanation in the current manuscript would provide useful context for interpreting the results.
2. Transgene Design and PEST Domains: The absence of functional PEST domains in the Notch3 fusion proteins is a critical design feature that is only made clear in the discussion. This detail should be introduced when the transgenic tools are first described in the Results section to ensure the reader fully understands the nature of the reporter lines from the beginning.
3. Spatial Statistics and 2D Limitations: The use of spatial statistics is elegant, well-executed and convincing. However, the analysis appears to be restricted to the apical layer of the neuroepithelium, effectively a 2D space. Please clarify whether this is a limitation inherent to the method. Does this approach exclude potential interactions with delaminating cells that may also express Notch ligands? Discussing this potential "vertical" signaling would make the proposed model more persuasive (see also below).

Minor Points

- Abstract (line 19): The phrase "positively interacts with Notch3" is imprecise and could be confusing. It does not explicitly state that this interaction leads to activation of Notch3 signaling. Please rephrase this to clarify the functional outcome of the interaction, e.g., "positively regulates Notch3 activity" or "activates Notch3 signaling."
- Figure S2A (page 2, lines 99-101): The current images are not adequate to support the claim that the transgenes have an identical expression pattern to endogenous Notch3 at 24 hours-post-fertilization. The whole-mount expression patterns shown do not have the resolution required for a meaningful comparison. Whole-mount images with more informative views are needed to compare global expression.
- Figure 1 Legend and Text (page 1, line 121): There is a discrepancy between the main text, Figure 1, and the figure legend. The text mentions GFAP, while the legend mentions ZO1, and the figure itself labels GFAP as Zrf1 (an anti-GFAP antibody). Please correct this by ensuring the figure label, legend, and text are consistent.
- TgBAC line description (page 3, lines 86-88): The sentence "and is intact at its endogenous notch3 locus" could be rephrased for better flow. A suggestion: "as it contains an intact endogenous Notch3 locus."
- Notch3 expression in NPs (page 3, lines 124-126): The inference that Notch3 might be active in neural progenitors (NPs) is not clearly supported by the preceding sentence describing the proportion of high and low nlsRFPpos cells. The authors should clarify the logic of this interpretation. If Notch3 expression in NPs has been previously characterized in the lab, a reference to that work would strengthen this point.
- Confusing Phrase (page 5, line 192): The phrase "productive gene copies of notch3" is confusing and should be clarified. It is not obvious what is meant by "productive."
- Page 6, lines 265–273: The authors argue that a simple lateral inhibition model cannot explain the observed Notch3 activity pattern. However, their analysis excludes *dla*-positive cells that are delaminating and may signal to NSCs. Could "vertical" signaling from these cells also contribute to the pattern? Discussing this possibility would strengthen the argument.
- Page 7, lines 295–: The description of Figure 5F,G may be incorrect. The panels appear swapped: the left panels likely correspond to control MO, while the right panels correspond to *jag1b* MO. This should be checked and corrected if needed.
- Page 7, lines 302–308: Upon *jag1b* knockdown, the number of Sox2+ cells does not change, but Sox2 levels are strongly reduced, while the salt-and-pepper expression of *dla* remains unchanged. This suggests that Sox2 expression depends mainly on *jag1b*/Notch3 signaling, with little contribution from *dla*. The authors should discuss this explicitly to refine their model.
- Hey1 Correlation (page 7, lines 312-327): The section on *hey1* expression and *jag1b*–Notch3 signaling is not fully convincing. The conclusion that Hey1 expression positively correlates with Jag1b (Fig. S7C,D) seems to contradict the figure legend's description of a "weak cell-autonomous correlation." (Spearman's $r = 0.38$). The authors should discuss this point and adjust the text to avoid over-interpreting the data.
- Cis-inhibition Sentence (page 9, lines 396-398): The sentence linking the reactivation frequency of *dla*-positive cells to cis-inhibition is not obvious for readers less familiar with the Notch pathway. It would be beneficial for the authors to either provide more context or consider removing the sentence to improve clarity.
- "Weak Results" (page 9, line 411): The phrase "weak results" in reference to unpublished experiments is vague and confusing. Please either describe these results in more detail or remove the sentence entirely, as it detracts from the clarity of the text.

(Remarks on code availability)

Reviewer #3

(Remarks to the Author)

In this manuscript, Ortica et al. address the mechanisms through which Notch signaling and two canonical Notch ligands, deltaA (dla) and jagged1b (jag1b), potentially interact to control the activity and state of adult neural stem cells in the telencephalon of zebrafish. The authors have generated and analyzed quantitatively Notch3-ICD fusion transgenic fish to assess the role of Notch3 and dla and jag1b in controlling neural stem cell activity. They extensively validate their various transgenic fish models and convincingly show that these report Notch3 expression and activity in neural stem/progenitor cells. Using knockdown and pharmacological inhibition of gamma-secretase, combined with quantitative microscopic analysis, they propose that Notch signaling plays a lateral inductive role in jag1b expression in the adult neurogenic cells where as Notch-dla is likely involved in a lateral inhibition signal. Further, they provide evidence that jag1b expression is necessary for regulating Notch3 activation in adult fish neural stem cells. Based on this and previous data from the same group, they propose that the combination of lateral induction and lateral inhibition signaling through Notch3 is responsible for maintaining neural stem cells in the fish telencephalon in a quiescent state. The authors have generated very important tools to study Notch3 signaling in vivo in zebrafish, and performed careful analysis of the activity of Notch3 and its putative ligands. These results and the tools will potentially be very important for the Notch and adult neurogenesis fields.

Minor comments

The authors use their Notch3-ICD fluoroprotein fusion transgenic fish to study neural stem cell activity and the roles of dla and jag1b. Although the authors have previously shown the importance of Notch3 in regulating quiescence in this system, the functional effects of their ligand manipulations could also involve other Notch proteins and other mechanisms. The authors should discuss these possibilities to provide a balanced picture.

The same minor critique is true for the gamma-secretase inhibition experiments. The authors use Notch3-ICD as a surrogate for Notch activation, but they mostly refer in the text to blocking Notch3. The authors should, in these global approaches, rather refer to reducing/blocking Notch signaling, as the approaches are not selective for Notch3. Changing the text to be more general will not detract from their findings.

(Remarks on code availability)

N/A

Version 1:

Reviewer comments:

Reviewer #1

(Remarks to the Author)

The authors should be congratulated for a lovely, rigorous study.

(Remarks on code availability)

Reviewer #2

(Remarks to the Author)

The authors have adequately addressed all the concerns raised in my previous report. I appreciate the effort put into the revision, which has clarified the points in question.

A minor point: in line 237, it should be "jag2b" instead of "jag2".

(Remarks on code availability)

Reviewer #3

(Remarks to the Author)

In their revised manuscript, Ortica et al. have addressed my minor concerns.

(Remarks on code availability)

Itemized response to Reviewers' comments

We would like to sincerely thank our three reviewers for their strong support and their very careful analysis of our work. Below in red is an itemized response to their comments, which truly helped strengthening and sharpening this manuscript.

Reviewer #1 (Remarks to the Author)

This manuscript uses expression of functional Notch3 fluorescent variants in BAC or knockin contexts to analyze the circuitry of Notch3 signaling in the zebrafish brain as it relates to neural stem cell quiescence. The authors leverage quantitative imaging, manipulation of Notch signals with inhibitors or Jag1b-directed morpholinos, and careful data analysis to draw the conclusion that Notch3 in NSCs is responding to Delta in neighboring cells, which results in the induction of Jag1b in the Notch3-high cells to maintain and reinforce the NSC fate in signal-receiving cells. The authors clearly indicate when their analyses provide robust results and when they show trends or outcomes that are not statistically significant. Overall, the work is thorough and support the authors' conclusions.

Additional comments

1. The authors note that previous modeling predicts that superposition of lateral induction on lateral inhibition results in more robust pattern formation. It would be interesting to include a mathematical model showing how the robustness of pattern formation varies as a function of the lateral induction signal strength.

→ Indeed, the relative strengths of lateral inhibition and lateral induction are the key parameters that determine the outcome of dual signaling. This is in fact already modelled in Mukherjee and Levine (2023) (referred to in our paper), where these parameters are encoded by the relative amounts of Delta and Jagged. This model predicts that increased robustness of pattern formation is observed at intermediate levels of Jagged-mediated signaling. This information has now been added to our manuscript (l.347-349).

2. The knockin of the fluorescent label onto Notch3 allows the authors to distinguish Notch3 signaling from signals due to activation of other Notch proteins in the same cell types. However, use of gamma secretase inhibitors results in the silencing of all Notch signals in treated cells. Can the authors comment on how the inhibition of all Notches might affect the interpretation of their results in Figs. 5A-5D?

→ It is correct that gamma-secretase inhibition is not selective of Notch3-mediated signaling. However, in the adult pallial germinal niche, *notch3* is the most highly expressed receptor gene in quiescent NSCs, *notch1* expression being enriched in activated cells. We are now documenting this directly by showing expression of the different notch genes on the NSPC UMAP (l.290-291 and new Fig.S6E), and also added the reference Alunni et al., 2023, where ISH profiles of *notch3* and *notch1b* are compared. We also toned down our conclusion statement to enlarge it to Notch (and not Notch3) (l.297).

3. The authors acknowledge the possibility that the half-life of the fluorescent fusions may differ from the wild-type Notch3 allele because the fusion sites alter or remove the PEST sequence implicated in protein degradation. How might longer persistence of the mutant N3ICD affect interpretation of the data? Is it possible to examine the difference in half-life of the wild-type and knockin N3ICD experimentally by monitoring the rate of decay of the two alleles after gamma secretase treatment? In the absence of an antibody detecting endogenous N3ICD, it is not possible to directly determine the extent to which the stability of N3ICD-GFP and N3ICD-AG may be affected. The publication by Fryer et al. 2004, now cited, shows that N3ICD stability in vitro is increased from 45 to 180 minutes in the absence of PEST-dependent degradation. In the adult NSC system, a longer half-life of knockin

alleles within this range would slightly delay novel NSC activation events in permissive conditions, and possibly slightly lengthen lineage progression. We certainly do not have the tools to detect such precision, but our phenotypical analyses did not detect significantly altered NSC behaviors. We now discuss the issue of N3ICD stability directly (Discussion I.376-383) and also made this clearer in the Results section (Results I.174-176) in response to Referee 2.

Reviewer's Comments:

Reviewer #2 (Remarks to the Author)

This manuscript by Sara Ortica and colleagues presents a detailed and quantitative study on how Notch3, in response to its DeltaA and Jagged1b ligands, controls lineage progression during neurogenesis in the zebrafish pallial brain. The authors have developed and validated a set of genetic tools that enable precise monitoring of Notch3 activity, which they used to dissect the spatial control of this signaling pathway. The innovative tools and rigorous statistical analysis together yield novel insights into how Notch signaling is patterned in the zebrafish brain. The manuscript also provides a careful characterization of *dla* and *jag1b* expression, followed by functional experiments that test their respective contributions to Notch3 regulation. These findings led the authors to propose a model in which lateral inhibition and lateral induction are integrated to fine-tune Notch3 activity, thereby influencing the balance between neural stem cell quiescence and activation. This model provides a new perspective on how Notch3 regulates neurogenesis in the zebrafish pallial brain and offers a framework that is likely to be relevant across vertebrate systems.

Overall, the manuscript is a strong contribution that combines technical innovation with biological insight. The work advances our understanding of neurogenesis in the vertebrate pallium and will be of broad interest to researchers studying neural stem cell regulation and Notch signaling.

The paper is well organized, although at times the text is difficult to follow due to insufficient explanation of certain concepts. Addressing the following points might substantially improve the manuscript's clarity and impact.

Major Points

1. Apical localization and apical size: In several analyses, the apical localization of cells or the size of their apical domain is treated as an important parameter. However, the functional significance of this parameter is never explained. A brief description of why apical localization/size is relevant for the analyses presented here would greatly help the reader. Although this was addressed in a previous publication from the lab, a short explanation in the current manuscript would provide useful context for interpreting the results.

Indeed, the apical plane and NSPC apical size are important parameters, and this for different reasons:

- Our previous work using intravital imaging showed that, although NSCs extend long basal processes into the parenchyma and are most likely connected to many other cells sub-apically, it is sufficient to consider their apical plane to readout spatially patterned events at population scale, e.g., their activation events (Dray et al., 2021). This is why we consider the apical plane of NSPCs for spatial statistics in the present work. This is now explained in the Results section (I.208-212).

- In addition, there is a correlation between apical surface area ("apical size") and NSC state and fate, whereby apical size decreases with lineage progression, and increases during quiescence as cells approach division (Mancini et al., 2023). This information is important in the present paper when linking N3ICD levels and NSC states. We now state it directly in the relevant parts of the Results section (I.259-261 and 266-267).

2. Transgene Design and PEST Domains: The absence of functional PEST domains in the Notch3 fusion

proteins is a critical design feature that is only made clear in the discussion. This detail should be introduced when the transgenic tools are first described in the Results section to ensure the reader fully understands the nature of the reporter lines from the beginning.

The information was actually present in the Results section (l.82-98) but we have now added clear discussions on this point, further in the Results section when describing quantifications of N3ICD (l.174-176) and in a full Discussion paragraph (l.376-383).

3. Spatial Statistics and 2D Limitations: The use of spatial statistics is elegant, well-executed and convincing. However, the analysis appears to be restricted to the apical layer of the neuroepithelium, effectively a 2D space. Please clarify whether this is a limitation inherent to the method.

This limitation is not inherent to the mathematical method, for which there are actually no limits in dimensionality. It was chosen based on (i) our imaging possibilities (we cannot precisely image the basal process from whole-mount apical views, making an analysis of neighborhoods very imprecise below the apical surface), (ii) the fact that there are no major variation in the z position of nuclei, such that projecting them to an apical 2D surface does not change their relative distances, and, most importantly, (iii) the fact that apical measurements appear sufficient to capture neighborhood interactions explaining the spatial NSC activation pattern (see point 1 above, and Dray et al., 2021).

We now explain the latter points in the Results section (l.208-212).

Does this approach exclude potential interactions with delaminating cells that may also express Notch ligands? Discussing this potential "vertical" signaling would make the proposed model more persuasive (see also below).

Delaminating cells indeed express DeltaA. In our approach, these cells are taken into account as long as they possess a detectable apical surface. This process can be very long, in the range of 10 days or more before this apical surface is fully closed (see Rosa et al., Development, 2024), and delaminating cells are considered for this entire duration. But certainly, interactions occurring at deeper parenchymal levels -and that would not have consequences visible at the apical surface- are ignored. We now stress this major point in Discussion (l.453-460).

Minor Points

- Abstract (line 19): The phrase "positively interacts with Notch3" is imprecise and could be confusing. It does not explicitly state that this interaction leads to activation of Notch3 signaling. Please rephrase this to clarify the functional outcome of the interaction, e.g., "positively regulates Notch3 activity" or "activates Notch3 signaling."

This has been modified.

- Figure S2A (page 2, lines 99-101): The current images are not adequate to support the claim that the transgenes have an identical expression pattern to endogenous Notch3 at 24 hours-post-fertilization. The whole-mount expression patterns shown do not have the resolution required for a meaningful comparison. Whole-mount images with more informative views are needed to compare global expression.

We have now added fluorescent ISH panels in one area at 24 hpf where *notch3* expression is restricted, namely the tail bud (expression is excluded from the notochord).

- Figure 1 Legend and Text (page 1, line 121): There is a discrepancy between the main text, Figure 1, and the figure legend. The text mentions GFAP, while the legend mentions ZO1, and the figure itself labels GFAP as Zrf1 (an anti-GFAP antibody). Please correct this by ensuring the figure label, legend, and text are consistent.

We apologize for this and have corrected the Figure and its legend to read Gfap everywhere, for clarity.

- TgBAC line description (page 3, lines 86-88): The sentence "and is intact at its endogenous notch3 locus" could be rephrased for better flow. A suggestion: "as it contains an intact endogenous Notch3 locus."

This has been modified.

- Notch3 expression in NPs (page 3, lines 124-126): The inference that Notch3 might be active in neural progenitors (NPs) is not clearly supported by the preceding sentence describing the proportion of high and low nlsRFP^{pos} cells. The authors should clarify the logic of this interpretation. If Notch3 expression in NPs has been previously characterized in the lab, a reference to that work would strengthen this point.

Indeed, the fact that low nlsRFP^{pos} cells were generally Gfap^{neg} was not mentioned, and was the logical link in the conclusion. We have added this information (l.124)

- Confusing Phrase (page 5, line 192): The phrase “productive gene copies of notch3” is confusing and should be clarified. It is not obvious what is meant by “productive.”

We meant “capable of producing a functional protein” (by opposition to *notch3*^{fh332}, which is a non-functional gene copy). We have now spelt this information in full (l.196).

- Page 6, lines 265–273: The authors argue that a simple lateral inhibition model cannot explain the observed Notch3 activity pattern. However, their analysis excludes *dla*-positive cells that are delaminating and may signal to NSCs. Could “vertical” signaling from these cells also contribute to the pattern? Discussing this possibility would strengthen the argument.

As mentioned in our response to point 3 above, *dla*^{pos} cells that are delaminating are not excluded from the analysis (but cells located deeper into the parenchyma certainly are, some of which could be *dla*^{pos}). This is now discussed (Discussion l.453-460)

- Page 7, lines 295–: The description of Figure 5F,G may be incorrect. The panels appear swapped: the left panels likely correspond to control MO, while the right panels correspond to *jag1b* MO. This should be checked and corrected if needed.

Indeed, we apologize for this mistake. We have corrected it.

- Page 7, lines 302–308: Upon *jag1b* knockdown, the number of Sox2+ cells does not change, but Sox2 levels are strongly reduced, while the salt-and-pepper expression of *dla* remains unchanged. This suggests that Sox2 expression depends mainly on *jag1b*/Notch3 signaling, with little contribution from *dla*. The authors should discuss this explicitly to refine their model.

Although it is clear that *Dla* plays a predominant role in quiescence control, we have not assessed the potential direct or indirect effects of *Dla* on stemness with the same precision (i.e., measurement of Sox2 levels), and therefore preferred not to increase our statement.

- Hey1 Correlation (page 7, lines 312-327): The section on *hey1* expression and *jag1b*–Notch3 signaling is not fully convincing. The conclusion that *Hey1* expression positively correlates with *Jag1b* (Fig. S7C,D) seems to contradict the figure legend's description of a “weak cell-autonomous correlation.” (Spearman's $r = 0.38$). The authors should discuss this point and adjust the text to avoid over-interpreting the data.

We have modified our text to mention a weak positive correlation of *hey1* transcripts with *jag1b* expression (Results l.331) and toned down our conclusion (“but *hey1* expression clearly also responds to other inputs », l.339).

- Cis-inhibition Sentence (page 9, lines 396-398): The sentence linking the reactivation frequency of *dla*-positive cells to cis-inhibition is not obvious for readers less familiar with the Notch pathway. It would be beneficial for the authors to either provide more context or consider removing the sentence to improve clarity.

We have added a line of explanation for cis-inhibition in this context (l.418).

- “Weak Results” (page 9, line 411): The phrase “weak results” in reference to unpublished experiments is vague and confusing. Please either describe these results in more detail or remove the sentence entirely, as it detracts from the clarity of the text.

We have deleted this sentence (l.431). What we observed experimentally when blocked *Jag1b* in a cell-autonomous manner (by electroporation of a tagged *jag1b*MO into adult NSCs in vivo) were very variable N3ICD levels in electroporated cells, which we also failed to interpret in link with immediate apical neighborhoods. These results may reveal more subtle cell interactions but that we have not deciphered at present.

Reviewer #3 (Remarks to the Author):

In this manuscript, Ortica et al. address the mechanisms through which Notch signaling and two canonical Notch ligands, deltaA (*dla*) and jagged1b (*jag1b*), potentially interact to control the activity and state of adult neural stem cells in the telencephalon of zebrafish. The authors have generated and analyzed quantitatively Notch3-ICD fusion transgenic fish to assess the role of Notch3 and *dla* and *jag1b* in controlling neural stem cell activity. They extensively validate their various transgenic fish models and convincingly show that these report Notch3 expression and activity in neural stem/progenitor cells. Using knockdown and pharmacological inhibition of gamma-secretase, combined with quantitative microscopic analysis, they propose that Notch signaling plays a lateral inductive role in *jag1b* expression in the adult neurogenic cells whereas Notch-*dla* is likely involved in a lateral inhibition signal. Further, they provide evidence that *jag1b* expression is necessary for regulating Notch3 activation in adult fish neural stem cells. Based on this and previous data from the same group, they propose that the combination of lateral induction and lateral inhibition signaling through Notch3 is responsible for maintaining neural stem cells in the fish telencephalon in a quiescent state. The authors have generated very important tools to study Notch3 signaling in vivo in zebrafish, and performed careful analysis of the activity of Notch3 and its putative ligands. These results and the tools will potentially be very important for the Notch and adult neurogenesis fields.

Minor comments

The authors use their Notch3-ICD fluoroprotein fusion transgenic fish to study neural stem cell activity and the roles of *dla* and *jag1b*. Although the authors have previously shown the importance of Notch3 in regulating quiescence in this system, the functional effects of their ligand manipulations could also involve other Notch proteins and other mechanisms. The authors should discuss these possibilities to provide a balanced picture.

This is true. We have now been careful in our conclusions on Jag1b/Notch3 signaling when jag1bMO phenotypes were analyzed (Results I.321 and 338) and added this point in the discussion (I.422-424).

The same minor critique is true for the gamma-secretase inhibition experiments. The authors use Notch3-ICD as a surrogate for Notch activation, but they mostly refer in the text to blocking Notch3. The authors should, in these global approaches, rather refer to reducing/blocking Notch signaling, as the approaches are not selective for Notch3. Changing the text to be more general will not detract from their findings.

We now also insist on this point (Results I.290-291).

Finally, in response to the two points above, we also added a supplementary figure panel (new Fig.S6E) documenting the expression profile of all notch genes on the scRNAseq UMAP of adult pallial NSPCs, showing the predominant expression of notch3.

Reviewer #3 (Remarks on code availability):

N/A